## OPEN

# Chromosome-scale and haplotype-resolved genome assembly of a tetraploid potato cultivar

Hequan Sun [1,2,8], Wen-Biao Jiao [1,2,3,4,8], Kristin Krause [2,7], José A. Campoy[2], Manish Goel [1,2], Kat Folz-Donahue[5], Christian Kukat [5], Bruno Huettel[6] and Korbinian Schneeberger [1,2] ✉

**Potato is the most widely produced tuber crop worldwide. However, reconstructing the four haplotypes of its autotetraploid genome remained an unsolved challenge. Here, we report the 3.1 Gb haplotype-resolved (at 99.6% precision), chromosome-scale assembly of the potato cultivar 'Otava' based on high-quality long reads, single-cell sequencing of 717 pollen genomes and Hi-C data. Unexpectedly, ~50% of the genome was identical-by-descent due to recent inbreeding, which was contrasted by highly abundant structural rearrangements involving ~20% of the genome. Among 38,214 genes, only 54% were present in all four haplotypes with an average of 3.2 copies per gene. Taking the leaf transcriptome as an example, 11% of the genes were differently expressed in at least one haplotype, where 25% of them were likely regulated through allele-specific DNA methylation. Our work sheds light on the recent breeding history of potato, the functional organization of its tetraploid genome and has the potential to strengthen the future of genomics-assisted breeding.**

Potato (*Solanum tuberosum*) is an important tuber crop and is among the five most produced crops in the world. Globally more than 350 billion kg of potato are produced per year with an increasing trend particularly in developing countries in Asia[1]. Despite the social and economic importance, the breeding success of potato remained low over the past decades due to its heterozygous, autotetraploid genome and the high levels of inbreeding depression, which challenge usual breeding strategies commonly applied to inbred, diploid crops[2,3].

A fundamental tool for modern breeding is the availability of reference sequences. The reference sequence for potato was generated from a double haploid plant, DM1-3 516 R44 (DM) and was initially published in 2011[4] and continuously improved over the past years including a recent update based on long read sequencing[5,6]. Another major advancement in potato genomics was the recent assembly of a heterozygous diploid potato, RH89-039-16 (RH)[7]. This haplotype-resolved genome was generated from a variety of different sequencing technologies and phase information from a genetic map derived from selfed progeny.

However, as of now, there is no haplotype-resolved assembly of a tetraploid potato cultivar available nor is there a straightforward method that would enable the assembly of the individual haplotypes of a tetraploid genome. The latest methods for haplotype phasing include the separation of sequencing reads based on the differences between the parental genomes[8] or on haplotype information derived from gamete[9-12] or offspring genomes[7,13]. Similarly, chromosome conformation capture sequencing (for example, Hi-C) can help to resolve haplotypes during or before the assembly[14-18] and has been applied to polyploids already[15-17]. However, even though straightforward in its application, chromosome conformation capture sequencing can lead to haplotype switch errors and requires additional efforts such as genetic maps for correction[7,9,18].

## Results

**Genome assembly of a tetraploid potato.** We generated an assembly of the autotetraploid genome of *S. tuberosum* 'Otava' using high-quality long PacBio HiFi reads (30× per haplotype) using hifiasm[19] (Fig. 1, Supplementary Table 1 and Extended Data Fig. 1; Methods). The initial assembly consisted of 6,366 contigs with an N50 of 2.1 megabases (Mb) (Supplementary Fig. 1). While the total assembly size of 2.2 gigabases (Gb) was much larger than the estimated haploid genome size of ~840 Mb, it accounted only for ~65% of the tetraploid genome size (Extended Data Fig. 1b) indicating that a major portion of the genome collapsed during the assembly. A sequencing depth histogram across the contigs featured four distinct peaks, which originated from regions with either one, two, three or four (collapsed) haplotype(s) (Fig. 1b). While most of the contigs represented only one haplotype (referred to as haplotigs) and accounted for 1.5 Gb (68%) of the assembly, contigs representing two, three or even four collapsed haplotypes (referred to as diplotigs, triplotigs or tetraplotigs) still made up 470 Mb (21%), 173 Mb (8%) or 43 Mb (2%). Regions with even higher coverages were virtually absent (9.4 Mb, 0.4%).

As there is no straightforward solution to untangle collapsed contigs after the assembly, we restarted the genome assembly but this time based on four separated read sets each derived from one of the four haplotypes. In diploids, such a read separation before the assembly can be performed by sorting the reads according to their similarity to the parental genomes (trio binning)[8]. But as autotetraploid individuals inherit two haplotypes through both the maternal and paternal lineages, this cannot be applied to autotetraploid genomes. Alternatively, the reads can also be separated using the haplotypes found in gamete genomes (gamete binning)[9]. While this is straightforward with haploid gametes from diploid individuals, tetraploid potato develops diploid gametes, which again does not separate individual haplotypes. However, as the pairing of the two

[1]Faculty of Biology, LMU Munich, Planegg-Martinsried, Germany. [2]Department of Chromosome Biology, Max Planck Institute for Plant Breeding Research, Cologne, Germany. [3]Key Laboratory of Horticultural Plant Biology (Ministry of Education), Huazhong Agricultural University, Wuhan, China. [4]College of Informatics, Huazhong Agricultural University, Wuhan, China. [5]FACS & Imaging Core Facility, Max Planck Institute for Biology of Ageing, Cologne, Germany. [6]Max Planck-Genome-center Cologne, Cologne, Germany. [7]Present address: Illumina Solutions Center Berlin, Berlin, Germany. [8]These authors contributed equally: Hequan Sun, Wen-Biao Jiao. ✉e-mail: k.schneeberger@lmu.de

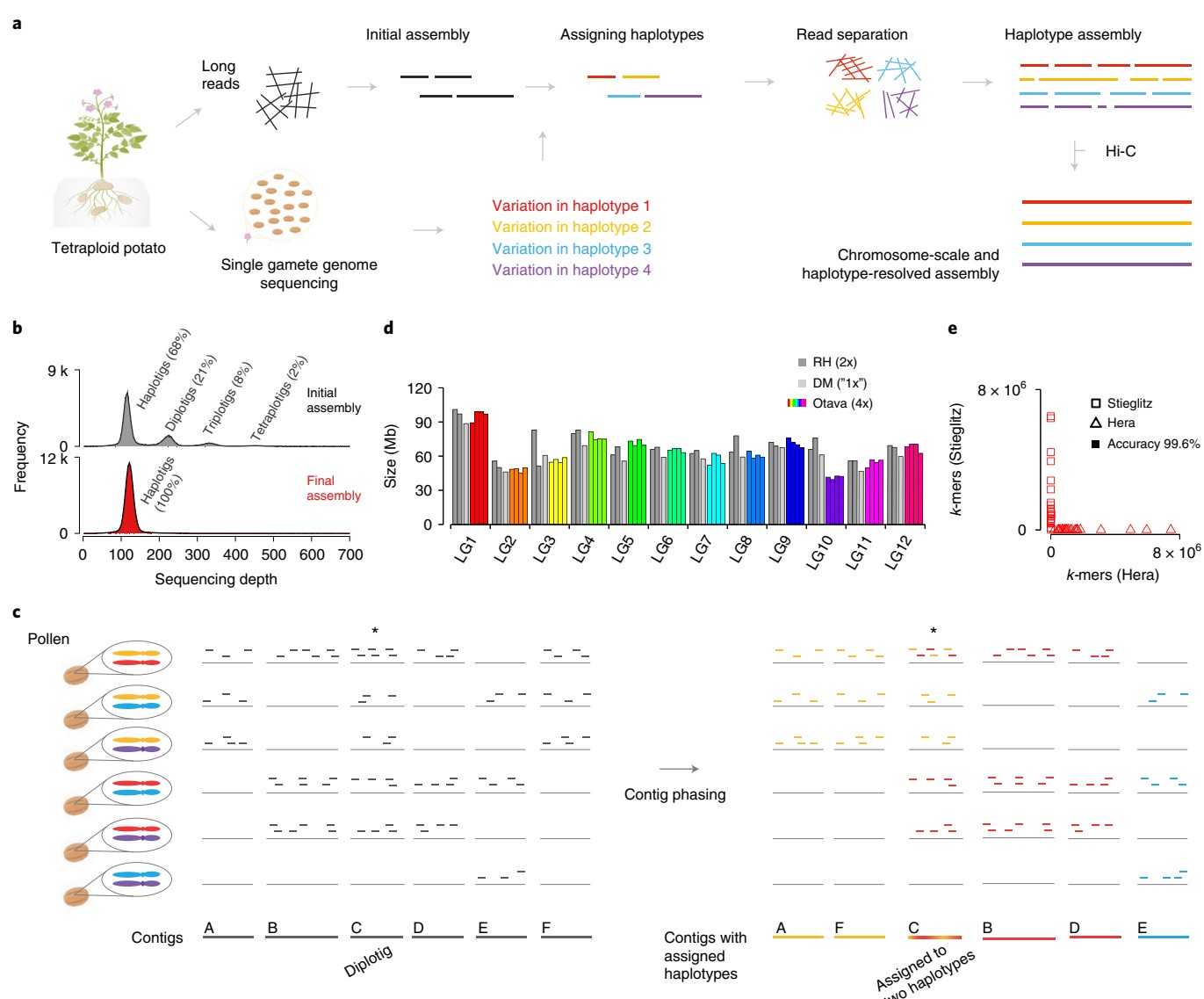

**Fig. 1 | Haplotype-resolved assembly of an autotetraploid potato genome. a**, Assembly strategy (gamete-binning) for tetraploid genomes. Long reads are sequenced from somatic DNA and an initial contig-level assembly is generated. In addition, sequencing data of gamete genomes are generated. Genetic linkage enables grouping of the contig into clusters, which represent the individual haplotypes. Long reads are assigned to haplotypes on the basis of their similarity to the contigs. Each haplotype can be assembled separately and scaffolded to chromosome-scale using Hi-C. The potato plant figure was created with BioRender.com. **b**, Histogram of sequencing depth within 10-kb windows of the initial assembly (top) revealed the presence of haplotigs (68.3%), diplotigs (21.4%), triplotigs (7.9%) and tetraplotigs (2.0%). As a comparison, only one major peak was observed (with increased frequency) in the final assembly (bottom). On the y-axis, 'k' represents for '1,000'. **c**, Linkage grouping. PAPs (presence/absence patterns) at each coverage marker (50-kb region) are defined by the absence or presence of corresponding sequencing reads from each of the pollen genomes. For instance, the PAP of contig A is '111000', where '1' refers to pollen genomes with reads that align to the contig and '0' refers to pollen genomes without such reads. PAPs of coverage markers A and F are highly correlated and can be grouped as {A, F}. Similarly, B and D are grouped as {B, D}, while E remains ungrouped as {E}. The diplotig coverage marker C shows highest correlations to {A, F} and {B, D} as compared to {E} and therefore extends these to larger clusters. The final result is the three clusters {A, C, F}, {B, C, D} and {E}. **d**, Assembly sizes of the haplotypes, which were highly consistent with the DM[5] and RH[7] assemblies. **e**, k-mer based evaluation of the haplotyping accuracy. Each point represents one individual haplotype of one chromosome. The values on the x and y axes indicate the numbers of k-mers within the haplotype sequence that are unique to either of the parental genomes of 'Hera' or 'Stieglitz'. Overall, 99.6% of the variation was correctly phased.

haplotypes in a diploid gamete is random in potato, we speculated that it might be possible to gain information on individual haplotypes (and thus to separate the reads into four distinct sets) if we sequence a sufficient number of diploid gametes.

To test if gamete binning could be applied for the genome assembly of 'Otava', we sequenced the genomes from 717 pollen nuclei with Illumina short reads with an average sequence coverage of 0.18× (Supplementary Fig. 2) and aligned each of the 717 read sets against the initial assembly. As defining a high-density single-nucleotide polymorphism (SNP) list can be difficult in a highly heterozygous autotetraploid genome, we defined 'coverage markers' (using average alignment depth in 50 kilobase (kb) windows) to assess if a genomic region was present in a pollen genome or not (Methods).

A coverage marker will be covered by reads if one of the two haplotypes of a pollen carries the region of the coverage marker. With this, we could assess the presence/absence pattern (PAP) of a coverage marker across all the 717 pollen genomes (Fig. 1c). Closely linked markers feature highly similar PAPs, however, recombination breakpoints, which are integrated into the pollen genomes during meiosis, change the haplotype within a pollen genome and thereby slightly change the PAPs along the chromosome. But, as recombination is generally rare, closely linked coverage markers still feature highly correlated PAPs. We therefore could use the similarities between the PAPs to cluster the contigs into 48 groups representing the four haplotypes of all 12 chromosomes (Extended Data Fig. 2 and Supplementary Fig. 3). Haplotigs were assigned to single clusters. Diplotigs, triplotigs and tetraplotigs represented multiple haplotypes and were therefore assigned to two, three or four of the clusters (Methods).

Once the contigs were assigned to haplotypes, the PacBio HiFi reads could also be assigned to these haplotypes on the basis of their alignments against the contigs. Reads aligned to diplotigs, triplotigs or tetraplotigs were randomly assigned to one of the respective haplotypes. With this, >99.9% of the nonorganellar PacBio HiFi reads could be assigned to one of the 48 read sets (Supplementary Fig. 4; Methods). Assembling the read sets using hifiasm resulted in 48 haplotype-resolved assemblies with an average N50 of 7.1 Mb and a total size of 3.1 Gb. Finally, we used Hi-C short read data (130× per haplotype) to scaffold the contigs of each assembly to a chromosome-scale, haplotype-resolved assembly (Extended Data Fig. 3; Methods). Comparison of the full assembly to whole genome sequencing short reads of 'Otava' using Merqury[20] revealed very high base accuracy (QV > 51.7) and completeness (97.3%) of the 'Otava' genome (Methods).

The sizes of the four haplotypes of each chromosome were highly consistent with each other as well as with those of the DM and RH assemblies[4–7] except for the consistently shorter assemblies of LG10, which indicated the presence of large-scale chromosomal rearrangements between different cultivars, similar to those previously described[21] (Fig. 1d). Apart from the LG10 differences, the 'Otava' assembly was in high synteny to the DM reference sequence suggesting that also the structure of the chromosomes was assembled correctly (Extended Data Fig. 4). To evaluate the haplotyping accuracy of the tetraploid assembly in more depth, we sequenced the parental cultivars of 'Otava', called 'Stieglitz' and 'Hera', with Illumina short reads at 10× coverage per haplotype, as each of the chromosomes was either inherited from 'Stieglitz' or from 'Hera'. Comparing the *k*-mers, which are specific to one of the parental genomes with each of the 48 chromosome assemblies, we found that each chromosome included almost exclusively *k*-mers from one but not the other parent revealing a haplotyping accuracy of 99.6% (Fig. 1e; Methods).

Integrating ab initio predictions, protein and RNA-seq read[4–7] alignments, we annotated 152,855 gene models across four haplotypes, with an overall benchmarking universal single-copy orthologs (BUSCO)[22] completeness score of 97.3%, which is highly comparable with the annotations of the RH and DM assemblies[5,7] (Supplementary Tables 2 and 3; Methods). In addition, we found comparable amounts of various different types of noncoding RNA for all haplotype genomes, which in total accounted for 33.9 Mb across the entire genome (Supplementary Table 4). Repetitive sequences made up 66% of the assembly with long terminal repeat retrotransposons as the most abundant class and ribosomal DNA clusters of up to 600 kb in size, which were assembled without any gaps (Supplementary Tables 5 and 6; Methods). The distribution of genes and repeats along the chromosome followed the typical distribution of monocentric plant genomes with high gene and low repeat densities at the distal parts of the chromosome, while in the pericentromeric regions the gene densities were low and the repeat densities were high (Fig. 2).

**The genomic footprints of inbreeding.** A histogram of sequence differences within 10-kb windows between the haplotypes revealed two separated peaks implying the presence of highly similar as well as highly different regions (Fig. 3a). The divergent regions included 1 SNP per 60 bp on average, while the remaining 50% of the regions were almost without any differences (Fig. 3a). This extreme similarity between some of the haplotypes suggested that they were recently inherited from a common ancestor. In fact, the pedigree of many of the cultivated potatoes, including 'Otava', contains cultivars that occur more than once in their ancestry[23,24] (Extended Data Fig. 1). Common ancestors in different lineages of the pedigree leads to inbreeding and results in regions that are identical-by-descent (IBD) between their haplotypes (Fig. 3b,c and Extended Data Figs. 5–10; Methods). The IBD blocks almost perfectly matched the collapsed regions in the initial assembly explaining the high degree of unresolved regions. To exclude any potential artefacts, we screened the IBD blocks for collapsed variation (using pseudoheterozygous variation in read alignments against the assembly) and found that a maximum of 1 SNP in 72 kb could have been missed (Supplementary Table 7), reassuring that highly similar IBD regions do exist in the genome.

Overall, almost 50% of the tetraploid genome of Otava was included in IBD blocks and was shared by either two, three or in rare cases even by four haplotypes (Fig. 3b–d). Individual IBD blocks varied in size and reached up to 41.6 Mb, while IBD blocks in the pericentromeres were significantly larger as compared to the IBD blocks in the distal parts of the chromosomes (Extended Data Figs. 5–10 and Supplementary Fig. 5). Even though it is possible that long IDB blocks were recently introduced and were not broken up by meiotic recombination yet, it is more likely that these extremely long IBD blocks exist due to local suppression of meiotic recombination in the pericentromeres (Fig. 3c). Using the accumulated mutation rates in the IBD blocks as an estimate of their age showed that long IBD blocks weren't younger as compared to short IBD blocks (Supplementary Fig. 5b).

**Extreme haplotype differences and their influence on genes.** The highly similar IBD blocks were contrasted by high levels of structural rearrangements between the nonshared regions of the genome (Fig. 3 and Extended Data Figs. 5–10; Methods). Inversions, duplications and translocations made up 3.8–42.9% of each of the haplotypes (or 19.3% of the genome) depending on the abundance of IDB blocks in the respective haplotypes. Duplications and translocations were highly enriched for Gypsy and Copia retrotransposons near their breakpoints revealing their active role during genome diversification as it was described for other plant genomes before[25], while inversions were not enriched for transposable elements (Supplementary Fig. 6 and Supplementary Table 8).

Excluding IBD blocks, structural rearrangements made up 15.0–65.8% of each chromosome. In addition, each haplotype included 11.0–42.5% of unique sequence that could not be aligned to any of the other haplotypes (Fig. 3d). This amount of structural variation and haplotype-specific sequence was much higher than has been reported for any other crop species so far, supporting earlier suggestions that genomic introgressions from wild relatives were part of the domestication history of potato[26].

Overall, we found 661 structural variations longer than 100 kb which all were supported by the contiguity of the assembled contigs or Hi-C contact signals, including 220 duplications, 207 translocations and 234 inversions (Fig. 2, Extended Data Figs. 3 and 5–10, Supplementary Fig. 7 and Supplementary Table 9). While comparable in number, inversions were much larger than the other types of rearrangements and reached sizes of up to 12.4 Mb (Fig. 3e,f). Although these large inversions were mostly located in the pericentromeric regions where genes occur at low density, they still harbored nearly 5% of all genes (7,958 out of 152,855). Meiotic

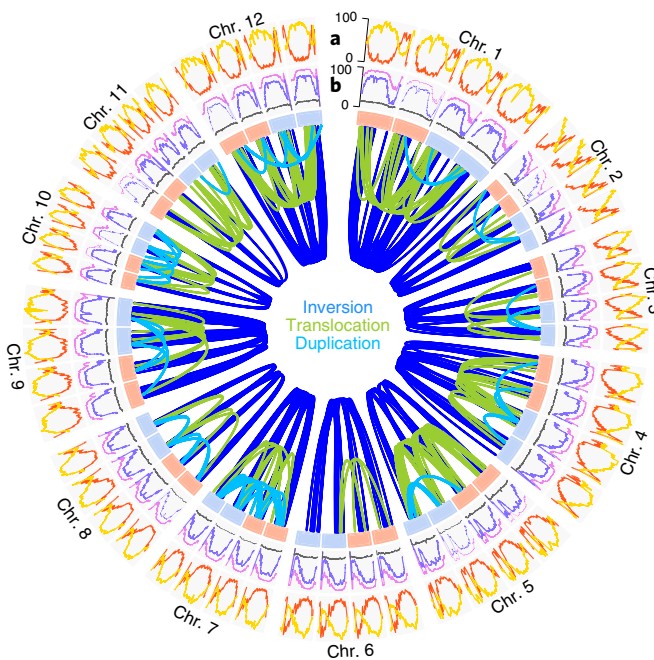

**Fig. 2 | The genomic features of the autotetraploid potato genome.**
**a**, Gene density (number of genes per Mb; red) and percentage of TE-related sequence (yellow) within 2-Mb windows along the four haplotypes of each of the 12 chromosomes (Chr. 1–12). **b**, Landscapes of methylation in CG (purple), CHG (blue), CHH (gray) context within 2-Mb windows. The links in the center show >600 structural rearrangements >100 kb found between the four haplotypes of each chromosome. Light blue and red boxes refer to the maternally and paternally inherited chromosomes.

crossover events from the pollen genomes were virtually absent in inversions, indicating that these regions are likely to introduce large segregating haplotypes among cultivated potato (Fig. 3c).

Pairwise allelic divergence of the genes ranged from 0 to 140 differences per kb and included identical as well as divergent alleles. The average pairwise difference of the divergent alleles was 18 differences per kb (Fig. 4a) and only 53.6% of the genes were present in all four haplotypes. The remaining 46.4% of the genes were present in three (20.0%), two (15.9%) or even only one (10.5%) of the haplotypes (Fig. 4b) with an average of 3.2 copies per gene. In addition, the coding sequences of some of these copies were identical to each other. For example, only 3,066 (15.4%) of the genes with four copies also featured four distinct alleles. In consequence, even though each gene featured 3.2 copies on average, there were only 1.9 distinct alleles per gene (Fig. 4b).

While it was expected to find identical gene alleles within the IBD blocks, only ~45% of the identical gene alleles were actually within shared regions. To test if the high number of identical alleles between the otherwise different haplotypes was indicative of selection, we tested whether these genes were enriched for specific functions. This revealed a significant enrichment for genes with gene ontology (GO) terms involving photosynthesis, chlorophyll binding and translation (Fig. 4c) suggesting a selection-induced reduction of allelic diversity through the optimization of plant performance.

The low number of distinct alleles per gene and a selection-induced reduction of allelic difference also implied that the tetraploid nature of the genome is not a necessary feature of the high performance of potato in different environments. However, transforming potato into a diploid suffers from the random distribution of the nonfunctional alleles throughout the individual haplotypes implying that

any ploidy reduction would lead to a significant gene loss. In fact, the BUSCO score (indicating completeness) of the annotations of the individual haplotypes was 89.5% on average, while the score of all four haplotypes combined was 97.3% (Supplementary Table 3) providing evidence that the individual haplotypes lack genes that are present elsewhere in the genome. Likewise, the doubled-monoploid DM[4,5] and the diploid RH[7] genomes feature 5,901 (15.4%) or 3,245 (8.5%), respectively, less genes as compared with the tetraploid genome. The gene family with highest percentage of genes with presence/absence variation (45.4%; 316 out of 696 genes) were the NLR resistance genes (Supplementary Table 10), which are known for their high intraspecies variability[27,28].

To investigate how genes are expressed in this tetraploid genome, we sequenced 367 million read pairs of the 'Otava' leaf transcriptome in three replicates (Supplementary Table 1; Methods). The four haplotype genomes contributed highly similar amounts of RNA, suggesting that none of the haplotypes was dominant (Fig. 4d and Supplementary Fig. 8), which is similar to observations in another autoploid species[15]. Comparable to earlier analyses on the effects of copy number variations (CNVs) on gene expression[29], the number of allelic copies also impacted on gene expression. Genes with more allelic copies showed a significantly increased gene expression as compared to genes with fewer allelic copies (Fig. 4e). Although gene expressions of the four haplotype genomes were comparable at genome scale, 10.9% of the genes with four allelic copies featured significant expression differences between the individual alleles, which were enriched in hydrolase activity, photosynthesis, light harvesting and RNA methylation (Fig. 4f and Supplementary Tables 11 and 12).

To understand more about the regulation of allele-specific expression, we sequenced the DNA methylome of 'Otava' using enzymatic methylome sequencing with three replicates, each with 277 million read pairs (Supplementary Table 1 and Methods). Overall, we found that DNA methylation was consistent across all haplotypes while DNA methylation levels in IBD blocks were slightly higher as compared to the nonshared regions in the other haplotypes (Fig. 2 and Supplementary Fig. 9). Of the 1,219 genes with significant differences in the allele-specific expression, 327 genes were significantly correlated with the level of methylation at the up/downstream regions of these genes (Fig. 4g, Supplementary Figs. 10 and 11 and Supplementary Tables 11–14), suggesting that ~25% of the allele-specific expression is regulated through DNA methylation.

## Discussion

Here, we reported the first high-quality haplotype-resolved assembly of an autotetraploid potato. Leveraging long reads and single-cell genotyping of diploid gametes, we were able to reconstruct the sequences of all four haplotypes. The structural rearrangements between the haplotypes were much higher as compared to the differences commonly found in natural populations and were rather reminiscent of the differences found between species. In fact, many of the potato cultivars contain genomic introgressions from wild species. In the pedigree of 'Otava', for example, we can find 'Edinense fraglich'. It was probably a variety of *S. demissum* or a hybrid of *S. demissum* and *S. edinense* or *S. tuberosum*, which was used to introduce resistance against *Phytophthora infestans* during the first decades of the last century[30].

The high level of sequence differences, however, was contrasted by widespread IBD blocks, which were most likely introduced by crossing related genotypes during breeding, even though we cannot exclude that some of these blocks might have been formed via double reduction during meiosis[31]. This similarity of the IBD blocks was the reason for the abundant collapsed regions in the initial assembly. As these regions were almost identical, it was not possible to assemble them from the sequence data alone. IBD blocks are

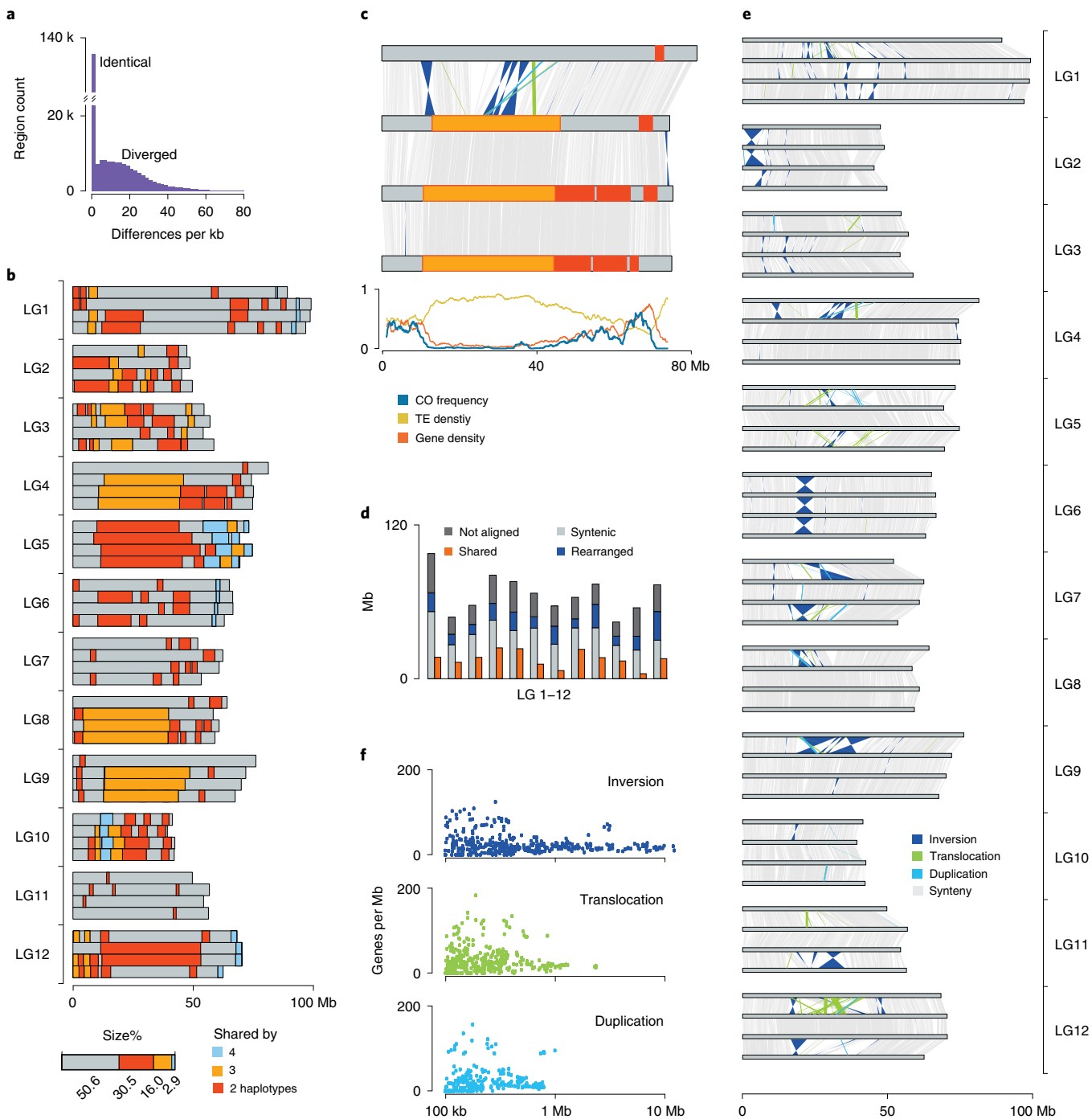

**Fig. 3 | In-depth haplotype analysis of the tetraploid genome. a**, SNP density as observed in pairwise comparisons between the haplotypes revealed two separated peaks. The high abundance of highly similar/identical regions suggested the existence of IBD blocks. On the y-axis, 'k' represents for '1,000'. **b**, IBD blocks (minimum size, 1 Mb) across the genome. Regions shared by two, three or four haplotypes are colored in red, orange or blue. **c**, A zoom-in on the IBD blocks and structural rearrangements of LG4. Large IBD blocks were more likely to occur in pericentromeric regions with low gene but high TE content and suppressed meiotic recombination. CO: crossing-over. Colors as defined in **b** and **e. d**, Average alignment statistics and structural rearrangements in each chromosome. **e**, Structural rearrangements between the four haplotypes of each chromosome. **f**, Correlation of the individual size of 220 duplications, 207 translocations and 234 inversions with the respective gene density.

a widespread phenomenon in many crops or livestock in general, though the challenges associated with the high similarity between haplotypes can be solved by using the power of genetics and analysing individual gamete genomes.

The abundance of IBD blocks, in addition to IBD-independent allele sharing, led to the unexpected observation that the

tetraploid genome only included 1.9 diverse allelic copies per gene. This implied that the maximal allelic diversity that could be included in the tetraploid genome was not reached, even though the high yield and yield stability of potato is supposed to be promoted by the effects of heterosis, which itself is based on nonadditive interactions of diverse alleles[32]. Whether the high abundance

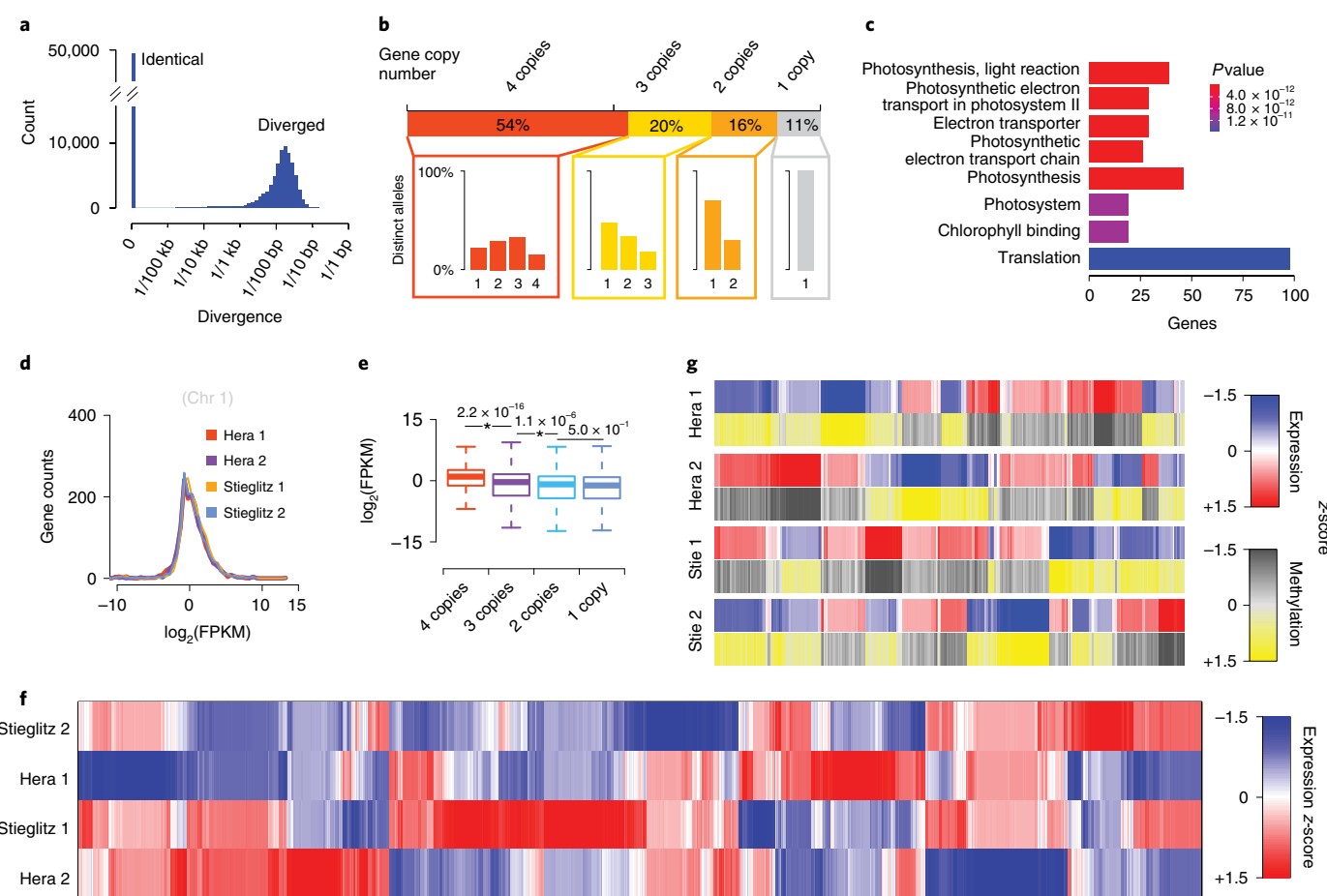

**Fig. 4 | Impact of haplotype divergence on genes and their expression. a**, Pairwise evaluation of allelic divergence of genes. **b**, Presence/absence variations of genes. Overall, 53.6%, 20.0%, 15.9% or 10.5% of the genes showed four, three, two or one allelic copies/copy within the tetraploid genome with an average of 3.2 allelic copies per gene. Different configurations of divergent alleles were observed for the sets of genes with four, three or two copies, for instance, within the genes with four allelic copies, one (22.3%), two (29.8%), three (32.5%) or four (15.4%) divergent allele/s were observed. Overall, this led to an average of 1.9 distinct alleles per gene. **c**, GO enrichment analysis of genes with four identical alleles ($P$ value adjusted with Benjamini–Hochberg method). **d**, Four haplotypes of chromosome 1 showed comparable amounts of expressed transcripts (FPKM; chromosomes 2–12 given by Supplementary Fig. 8). **e**, Genes with more allelic copies are with significantly higher expression than those with fewer copies ('*' indicates statistical significance: two-sided Student's $t$-test). Respective number of genes (satisfying $\log_2$-scaled FPKM over –15) with four copies—16,036, three copies—4,940, two copies—3,520 and one copy—1,906. Intervals for boxplots: center, median (50th percentile); lower bounds of box, 25th percentile (Q1); upper bounds of box, 75th percentile (Q3); lower whisker, maximum of (minima, Q1 – 1.5× IQR); upper whisker, minimum of (maxima, Q3 + 1.5× IQR). IQR, interquartile range (range of Q1 to Q3). **f**, Among the four haplotypes, 10.8% (1,219) of the 11,154 genes with four functional alleles (at least three samples showing counts per million reads >1.0) showed allele-specific expression (Supplementary Tables 11 and 12). **g**, Among those, allele-specific expression of 327 genes was significantly correlated with DNA methylation levels, which were measured in the 1-kb upstream or downstream regions surrounding the genes (two-sided correlation test; Supplementary Tables 13 and 14 and Supplementary Fig. 11).

of shared alleles suggests that the effects of heterosis could still be optimized by increasing the number of polymorphic alleles or if this indicates that the limits of heterosis were already reached remains to be seen.

Over the past years, considerable success has been made in redomesticating potato from a clonally propagated, tetraploid crop into a seed-propagated, diploid crop to increase reproduction rate, decrease costs in storage and transportation and improve disease control[2,33–35]. However, the random distribution of loss-of-function alleles in tetraploid potato can lead to the accelerated manifestation of inbreeding depression in the diploid genomes, when they are derived from tetraploids[7,36]. Haplotype-resolved assemblies of autotetraploids like the one presented here have the potential to support the design of optimal haplotypes by avoiding the combination of known incompatibility alleles[37]. Of course, this new possibility to assemble autotetraploid genomes does not eliminate all

breeding-related problems that result from the tetraploid nature of potato. However, being able to reconstruct the four haplotypes of cultivated potato is a breakthrough for modern genomics-assisted breeding strategies and ultimately has the power to increase the breeding success of potato in the future.

## Online content

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

## Methods

**Plant material.** Plant material was grown at Max Planck Institute for Plant Breeding Research (MPIPZ, Germany). The three potato cultivars 'Otava', 'Hera' and 'Stieglitz' (Extended Data Fig. 1) were clonally propagated and grown on Murashige-Skoog medium for 3–4 weeks at MPIPZ. Seedlings were transferred to soil in $7 \times 7\,\mathrm{cm}^2$ pots and grown in a Percival growth chamber for 2–3 weeks. Afterwards, potato plants were transferred to 1-l pots and grown until flowering. Potatoes were grown in long day conditions (16 h light, 8 h night cycle) at 22 °C. Details of DNA/RNA library preparation and sequencing are given in the Supplementary Information.

**Genome size estimation.** After trimming off 10× genomics barcodes and hexamers from the 370.3-Gb reads combined from the 10× single-cell CNV libraries and single-molecule libraries, $k$-mer counting ($k = 21$) was performed with Jellyfish (v.2.2.10)[38]. The $k$-mer histogram was provided to findGSE (v.1.0)[39] to estimate the haploid/tetraploid genome size of 'Otava' under the heterozygous mode (with 'exp_hom = 200'; Extended Data Fig. 1).

**Initial tetraploid genome assembly, polishing and purging.** The initial assembly of the tetraploid genome was performed using hifiasm (v.0.7)[19] with default settings with the 102.2-Gb raw PacBio HiFi reads of 'Otava', where the output consisting of unitigs (locally haplotype-resolved contigs) was selected for further processing. Then the short reads from the two 10× single-cell CNV libraries were aligned to the assembly using bowtie2 (v.2.2.8)[40]. These alignments were used to polish the assembly with pilon (v.1.22)[41] with options of --fix bases --changes --diploid --mindepth 0.8. Further, short reads of the two 10× single-cell CNV libraries, additional 10× single-molecule libraries and PacBio HiFi reads were all aligned to the polished assembly (using bowtie2 and minimap2 (v.2.17-r491)[42] respectively). Among 17,153 raw contigs, 9,041 contigs which were longer than 50 kb and with an average sequencing depth over 80× were kept. HiFi reads were re-aligned to the purged assembly and contigs covered less than 3× were removed. The HiFi reads-based purging process was repeated for five rounds to get an initial assembly of 6,366 contigs for subsequent analysis (Supplementary Fig. 1).

**Coverage marker definition by sequencing depth analysis.** All Illumina paired-end short reads from 10× single-cell/molecule libraries (and PacBio HiFi long reads) were aligned to the initial assembly of 6,366 contigs respectively using bowtie2 and minimap2. Potentially duplicated short reads were removed using picard MarkDuplicates function (http://broadinstitute.github.io/picard/). The depth along each contig was calculated with samtools depth function (v.1.9)[43] for each type of (short and long) reads and the depth at each position of each contig was taken as the sum of all types. Each contig was dissected into 10-kb windows and the average sequencing depth per base was calculated within each window. According to the average genome-wide depth of 113× per haplotype (denoted by $H$), windows with depths in [0, 170×], [171×, 283×], [284×, 396×], [397×, 509×] and [≥510×] were respectively determined as contig types of haplotig, diplotig, triplotig, tetraplotig and replotig, where the upper bound on depth for each type (if any) was determined by $H \times (i + 1/2)$, with $i$ being 1, 2, 3, 4. Neighboring 10-kb windows (along the same contig) were further merged as larger 50-kb coverage markers if they were classified as the same contig type (to ensure sufficient read signal for genotyping within each single pollen genome).

**Linkage-based grouping of contigs.** The read count (denoted by $r$) at each coverage marker of size $W$ (in bp) for each of the selected 717 pollen genomes (Supplementary Fig. 2), which were collected using bedtools (v.2.29.0)[44], was firstly normalized using $n_r = r \times (10^4/W) \times (10^6/N)$, where $N$ was the total number of reads aligned to the assembly of 6,366 contigs. Meanwhile, the average read count, $m_r$, for all coverage markers with more than $7 \times (N/10^6)$ reads was calculated for each pollen genome. The coverage (genotype) at a marker for each pollen was set to $n_r/m_r$ (which would be rounded to 0, 1 or 2). In general, coverages across 717 pollen read sets at the same window marker were used to build up a PAP: $X = x_1 x_2 \ldots x_{717}$, where $x_i$ was in {0, 1}, {0, 1, 2}, {1, 2} or {2}, depending on whether the marker type was haplotig, diplotig, triplotig or tetraplotig. The correlation between two PAPs $X$ and $Y$ was calculated as $\mathrm{cor}_{XY} = \mathrm{sum}[(x_i - m_x) \times (y_i - m_y)]$ / $[\mathrm{sqrt}(\mathrm{sum}(x_i - m_x)^2) \times \mathrm{sqrt}(\mathrm{sum}(y_i - m_y)^2)]$, where $m_x$ and $m_y$ were the respective average values. Initially, any pair of haplotigs/vertices (with sizes ≥100 kb) was connected by an edge, if the highest correlation value between the PAPs of the markers at two ends of the two haplotigs was >0.55 (Supplementary Fig. 3). This graph-based clustering[45] led to 48 groups representing the 48 haplotypes (Extended Data Fig. 2). If the lowest correlation value between any pair of the markers of any two groups was less than −0.25, they could be determined as homologous linkage groups (same chromosome, different haplotypes). With this, the 48 groups were clustered into 12 chromosomes, each with four different haplotypes.

Each of the remaining haplotigs $h$ (<100 kb) was integrated into the group with the marker showing the highest correlation with $h$. For a diplotig marker (with PAP $Z$) which represent the collapsed haplotypes $A_1$ and $A_2$, it can be expected that $Z \approx X + Y$, where $X$ and $Y$ are PAPs of two markers which closely linked to the diploid marker (Fig. 1c). We can therefore expect $Z\&X \approx X$ and $Z\&Y \approx Y$, where

'&' refers to the bit-wise AND-operation. As a result, the correlations of '$Z\&X$ with $X$' and '$Z\&Y$ with $Y$' should give the two highest values. Therefore, the two coverage markers from two of the 48 groups that show the highest correlations to a diplotig marker reveal the two groups that the diplotig will be assigned to. Similarly, triplotig markers can be associated with three groups. If all coverage markers of a contig are tetraplotig-type, the contig cannot be associated with any group because the information for linking a chromosome is missing. Only if at least one nontetraplotig coverage marker can be grouped, the tetraplotig marker of the contigs can be grouped.

Note, meiotic recombination can influence the linkage grouping. In the most simple case, that is, when no pollen genome carries a single meiotic recombination event, all PAPs at coverage markers from the same haplotype would be identical while PAPs at coverage markers from other haplotypes would be different (because haplotypes randomly occur in pollen genomes in a pairwise manner) and thus they can be easily grouped into haplotypes. In the presence of meiotic recombination, a few crossovers along each chromosome change the PAP values of the coverage markers but, as recombination is rare, the PAP values change only marginally. Thus, linkage grouping based on PAPs works even in the presence of meiotic recombination.

**Haplotype-specific PacBio HiFi read separation and haplotype assembly.** HiFi reads were classified into 48 groups on the basis of alignments to the 50-kb coverage markers using customized code[45]. Specifically, to assign a read to a marker, at least 500 bp of the read had to be aligned to the marker (reads aligning two neighboring markers were assigned to the marker with a larger overlapping size). Reads overlapping non-haplotig marker were randomly assigned to one of the marker-associated groups.

Each set of HiFi reads was independently assembled using hifiasm (v.0.7) with default settings. The resulting contigs were first polished with short reads using pilon with --fix bases --changes --diploid --mindepth 0.8 and then with HiFi reads using racon (v.1.4.10)[46] with -u --no-trimming.

**Evaluation of haplotyping accuracy.** For each haplotype assembly and the sequencing data of the parental genomes, $k$-mers ($k = 21$) were counted using KMC[47]. Specifically, $k$-mers found in 'Hera' but not in 'Stieglitz' (with a coverage of 6–12), as well as $k$-mers found in 'Stieglitz' but not in 'Hera' (with a coverage of 5–11) were selected using kmc_tools simple. For each haplotype, the sets of assembled $k$-mers were intersected with the two sets of parental-specific $k$-mers (using kmc_tools simple with subfunction intersect), which revealed $k$-mers common with either of the parental genomes. As a haplotype can only be inherited from one of the parents, it is expected to find parental-specific $k$-mers only of one parent. The overall haplotyping precision was determined as the total number of correctly phased $k$-mers divided by the total number of $k$-mers investigated in the 48 haplotype assemblies. Note, this was done before and after contig polishing, where we observed the same haplotyping accuracy.

**Haplotype-specific contig scaffolding using group-specific Hi-C reads.** Each haplotype-specific contig-level assembly was indexed with bwa index (with -a bwtsw) (v.0.7.15-r1140)[48] and samtools faidx. The haplotype-specific Hi-C read pairs were aligned using bwa aln and bwa sampe. Aligned reads (in pairs) were converted into BAM files using samtools view with options of -b -F12. The BAM files were filtered with filterBAM_forHiC.pl (from ALLHiC package, v.0.9.13)[49] to remove nonuniquely mapped reads. Then BAM files were converted to bed files using bamToBed (from bedtools package) and sorted by read name. The bed files were provided to *SALSA2* (run with -s 100000000 -m yes -i 10 -e DNASE)[50]. Potential chimeric contigs were broken at the chimeric sites given by SALSA2 output file of input_breaks, leading to a new set of contigs for each of the 48 original groups.

For each new group of contigs, the above process of contig indexing, Hi-C read alignment and BAM filtering was repeated. Then, for each haplotype, ALLHiC_partition was run with -e GATC -k 1 -m 25; allhic extract was run with --RE GATC; allhic optimize and ALLHiC_build were run with default settings; the chromosome contact map was visualized with ALLHiC_plot at 1-Mb resolution, where obvious mis-placement/orientation of large contigs were visually identified and manually corrected (Extended Data Fig. 3).

**Genome annotation and assessment.** Protein-coding genes for each haplotype chromosome were annotated with three types of evidences, including ab initio gene predictions (considering outputs by Augustus[51], GlimmerHMM[52] and SNAP[53]), transcripts assembled from Illumina short RNA-seq reads and alignments of homologous protein sequences. Specifically, protein sequences from *Solanum tuberosum* L.[5,7], *Arabidopsis thaliana* and other plant proteins from UnitProtKB were aligned to each haplotype assembly independently using Exonerate[54] with options of --percent 60 --minintron 10 --maxintron 60000. RNA-seq reads from two recent potato genome assembly work[5,7] were downloaded. All reads were first aligned to the full haplotype-resolved genome assembly using HISAT2 (v.2.2.0)[55]. Then for each of the 12 linkage groups, reads aligned to the respective four haplotypes were extracted and combined as one set with samtools view -L and bedtools bamtofastq. Within each linkage group, each set of reads from the

group was re-aligned to each haplotype sequence independently using HISAT2 and transcripts were assembled using StringTie[56]. Finally, all the above evidences were integrated with EvidenceModeler[57] to generate high-quality gene models for each haplotype assembly. Transposon elements (TE) were annotated using RepeatModeler and RepeatMasker (http://www.repeatmasker.org). TE-related genes were filtered by investigating their overlapping with TEs (overlapping percentage >30%), sequence alignment with TE-related protein sequences and *A. thaliana* TE-related gene sequences (requiring blastn alignment identity and coverage >30%).

To rescue potentially mis-annotated genes, all gene models were further improved. Specifically, against each of the four haplotypes, we first aligned the gene sequences of the other three haplotypes using blastn and, similarly, aligned the protein sequences of the other three haplotypes against those from the target haplotype using blastp[58] and Scipio[59]. If there were counterparts in the target haplotype to the other three (based on the alignments), the potential missing genes were added according to gene models given by ab initio prediction. Besides, gene models were split into smaller genes or merged as larger genes if all the alignments of genes from the other three haplotypes indicated a mis-merged or mis-split gene model.

The final assembly and annotation completeness were evaluated by BUSCO (v.4.1.4)[60] with 2,326 single-copy genes from the lineage database eudicots_odb10. The functional annotation of genes was performed with InterProScan (v.5.48)[61] with default parameters except for option -goterms. The GO terms were extracted for GO functional enrichment analysis by the R package ClusterProfiler[62]. The noncoding RNA was annotated with the tool Infernal (v.1.1)[63] by searching the database Rfam (v.14.3)[64]. The adjacent rDNAs with distance <5 kb were clustered as the potential rDNA clusters.

**Comparison of chromosome sequences.** Within each of the 12 homologous linkage groups (LGs), the chromosome-level sequences of the four haplotypes were aligned to each other as well as to the recently assembled DM genome using minimap2 with -ax asm20 --eqx. For each pair of haplotypes, the alignments were provided to SyRI[65], which searched for synteny, single-nucleotide level differences as well as large-scale structural variations (with -k -F S).

**Allelic expression analysis.** *Quality control.* Short reads from RNA sequencing were trimmed with Trimmomatic (v.0.39)[66] under paired-end mode, with options ILLUMINA:adapters.fa:2:30:10:8:true (adapters provided by the tool itself) SLIDINGWINDOW:4:15 LEADING:3 TRAILING:3 MINLEN:36.

*Read separation.* All reads were aligned to the final haplotype-resolved assembly using HISAT2 (v.2.2.0) with option -k 1 and reads were thus separated into 48 haplotype-specific groups with samtools view -L and bedtools bamtofastq.

Expression analysis was performed following the literature[67], using Stieglitz-1 genome as reference. Within each of the 12 linkage groups, the four sets of haplotype-wise RNA-seq reads were independently aligned to the respective Stieglitz-1 chromosome as reference, using HISAT2. The number of fragments at each gene from each chromosome was quantified using HTSeq (v.0.13.5)[68] with options --mode=union --nonunique=none --secondary-alignments=ignore --stranded=no (given the gff file for that chromosome). For testing dominance in allelic expression and effect of allelic copy number on expression, the fragment counts were normalized as fragments per kilobase per million reads (FPKM) and log2-scaled. The genes with four allelic copies were selected in analysing differential allele expression, where the log2-transformed CPM (count per million reads) at each gene was used for measuring the allele expression level. After excluding genes with less than three replicates showing CPM > 1.0, 11,154 expressed genes were kept. Paired comparisons on expression of the four alleles were performed and 1,219 were found to be differentially expressed in at least one pair of haplotypes with $P < 0.05$ adjusted using Benjamini–Hochberg method (Supplementary Table 11).

Note that the well clustering of the three biological replicates regarding four haplotypes based on the haplotype-specific expression of all genes from Stieglitz-1 genome (using hclust in R) showed that there was a high consistency between the replicates (Supplementary Fig. 10), thus the final FPKM value at each gene was taken as the average of the values of the three replicates in all related analysis (except for the procedure of differential analysis where replicates were not merged). At a gene, for four alleles with FPKM values of $x = \{x1, x2, x3, x4\}$, the $z$-score (in heatmaps) for each allele $i$ (in $\{1,2,3,4\}$) was calculated as $(xi - \text{mean}(x))/ \text{s.d.}(x)$, with mean($x$) giving the average of $x$ and s.d.($x$) giving the standard deviation of $x$.

**Methylation analysis.** *Quality control.* Short reads from methylation sequencing were trimmed with Trimmomatic (v.0.39) under paired-end mode, with options ILLUMINA:adapters.fa:2:30:10:8:true (adapters provided by the tool itself) SLIDINGWINDOW:4:15 LEADING:3 TRAILING:3 MINLEN:36.

*Read separation.* All reads were first aligned to the initial assembly (of 6,366 contigs) using bismark (v.0.23.0)[69] (with options --hisat2 --score_min L,0,-0.6), to avoid the default masking of reads from IBD regions. If a read pair could be aligned to coverage markers linking a single haplotype group, then it was assigned to that

group. If a read pair could be aligned to coverage markers linking multiple groups, it was randomly assigned to one of the groups.

*Methylation calling at 48 haplotype-specific chromosomes.* The reads assigned to each haplotype-specific group were independently re-aligned to the respective haplotype-resolved chromosome using bismark under the same options plus --non_directional (which gave alignments to all four bisulfite strands). Each bam file was deduplicated using deduplicate_bismark, after which methylation at CG, CHG and CHH contexts were simultaneously called with bismark_methylation_ extractor with options --comprehensive --cytosine_report --CX --bedGraph. Given a specific genomic region/window, the methylation level at CG(/CHG/CHH) context was determined as C_count/(C_count + T_count), where C_count was the number of reads carrying the methylated cytosine in CG(/CHG/CHH) context and T_count was the number of reads carrying un-methylated cytosine in CG(/CHG/CHH) context. Following this, a sliding window-based quantification of the methylation level was performed along the 48 chromosomes (window size, 2 Mb; step, 50 kb; Fig. 2b).

*Methylation calling at 12 Stieglitz-1 chromosomes as references.* Within each of the linkage groups, the four haplotype-wise reads were independently aligned to the respective Stieglitz-1 chromosome as reference using bismark under the same options (as above) plus --non_directional (which gave alignments to all four bisulfite strands). The methylation calling was done in a similar way as given above. The results were used to perform correlation analysis with allele-specific expression (Fig. 4g).

*Comparison of the level of methylation between IBD blocks and the corresponding regions in synteny in homologous haplotypes.* IBD blocks (at 50-kb resolution) that were shared by two or three haplotypes were investigated. According to the synteny between haplotypes (from SyRI-based analysis), the counterparts of such IBDs were located in the haplotypes being homologous to the haplotypes showing the IBDs. The counterparts were re-assigned to IBD groups if they were harbored by any known IBD blocks. This led to two sets, one with IBD blocks and the other with regions from other haplotypes but in synteny with the IBD blocks. Methylation level at CG(/CHG/CHH) context was calculated for each block within the two sets and $t$-test was used to investigate the difference between the two sets (Supplementary Fig. 9).

Note that, using the 12 Stieglitz-1 chromosomes as reference, the well clustering of the three biological replicates regarding four haplotypes based on CG, CHG or CHH methylation levels in 50-kb windows at a step of 25-kb (using hclust in R) showed that there was a high consistency between the replicates (Supplementary Fig. 10), thus the final methylation level (and the related $z$-score, similarly calculated as given in gene expression analysis) at each window was taken as the average of the levels at three replicates in all related analysis.

Other methods used in analysis have been provided in the Supplementary Information.

**Statistics.** All presented $P$ values correspond to two-sided $P$ values. Correlation test between the levels of allele-specific expression and methylation was done using cor. test function in $R$.

**Reporting Summary.** Further information on research design is available in the Nature Research Reporting Summary linked to this article.

## Data availability

High-throughput sequencing data analysed in this project are available under NCBI BioProject PRJNA751899. This Whole Genome Shotgun project (including assembly and annotation) has been deposited at DDBJ/ENA/GenBank under the accessions JAIVGA000000000, JAIVGB000000000, JAIVGC000000000 and JAIVGD000000000. The version described in this paper is version JAIVGA010000000, JAIVGB010000000, JAIVGC010000000 and JAIVGD010000000. The genome assembly and gene annotation of Otava are also available on Spud DB (http://spuddb.uga.edu/)[6]. Source data are provided at Zenodo (https://doi.org/10.5281/zenodo.5796752)[70]. Genome information for DM and RH used in this study are available on Spud DB (http://spuddb.uga.edu/).

## Code availability

Custom code and scripts supporting this work are available at github.com/schneebergerlab/GameteBinning_tetraploid or Zenodo (https://doi.org/10.5281/zenodo.5775114)[45].

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

## Acknowledgements

We would like to thank H. van Eck (Wageningen University & Research (WUR), the Netherlands), C. Gebhardt (Max Planck Institute for Plant Breeding Research (MPIPZ), Germany), Klaus J. Dehmer (Leibniz Institute of Plant Genetics and Crop Plant Research (IPK), Germany) and Benjamin Stich (Heinrich Heine University, Germany) for helpful discussions. We thank B. Walkemeier and C. Sänger (both MPIPZ) for plant cultivation and C. Brandt and K. J. Dehmer (both IPK) for providing material. We also thank P. J. Flood (Infarm, the Netherlands) for comments on the manuscript, S. Pophaly (MPIPZ) for help in data management and the three reviewers for valuable suggestions on the manuscript. We are very grateful to J. Hamilton and C. R. Buell (both University of Georgia, USA) for integrating the 'Otava' assembly into Spud DB (http://spuddb.uga.edu/)[6]. This work was funded by Deutsche Forschungsgemeinschaft (DFG, German Research Foundation) under Germany´s Excellence Strategy—EXC 2048/1—390686111 (K.S.) and European Research Council (ERC) grant 'INTERACT' (802629) (K.S.), the Humboldt Research Fellowship for 'Experienced Researchers' (Alexander von Humboldt Foundation) (J.A.C.) and the Marie Skłodowska-Curie Individual Fellowship PrunMut (789673) (J.A.C.).

## Author contributions

H.S., J.A.C. and K.S. conceptualized the project. K.S. supervised the project. H.S. designed and implemented the genome assembly pipeline, performed data analysis and drafted the manuscript. H.S., K.K., J.A.C., K.F-D., C.K. and B.H. generated the data. W-B.J. and M.G. performed data analysis. H.S. and K.S. finalized the manuscript with input from all authors. All authors read and approved the final manuscript.

## Funding

## Competing interests

The authors declare no competing interests.

## Additional information

**Extended data** is available for this paper at https://doi.org/10.1038/s41588-022-01015-0.

**Correspondence and requests for materials** should be addressed to Korbinian Schneeberger.

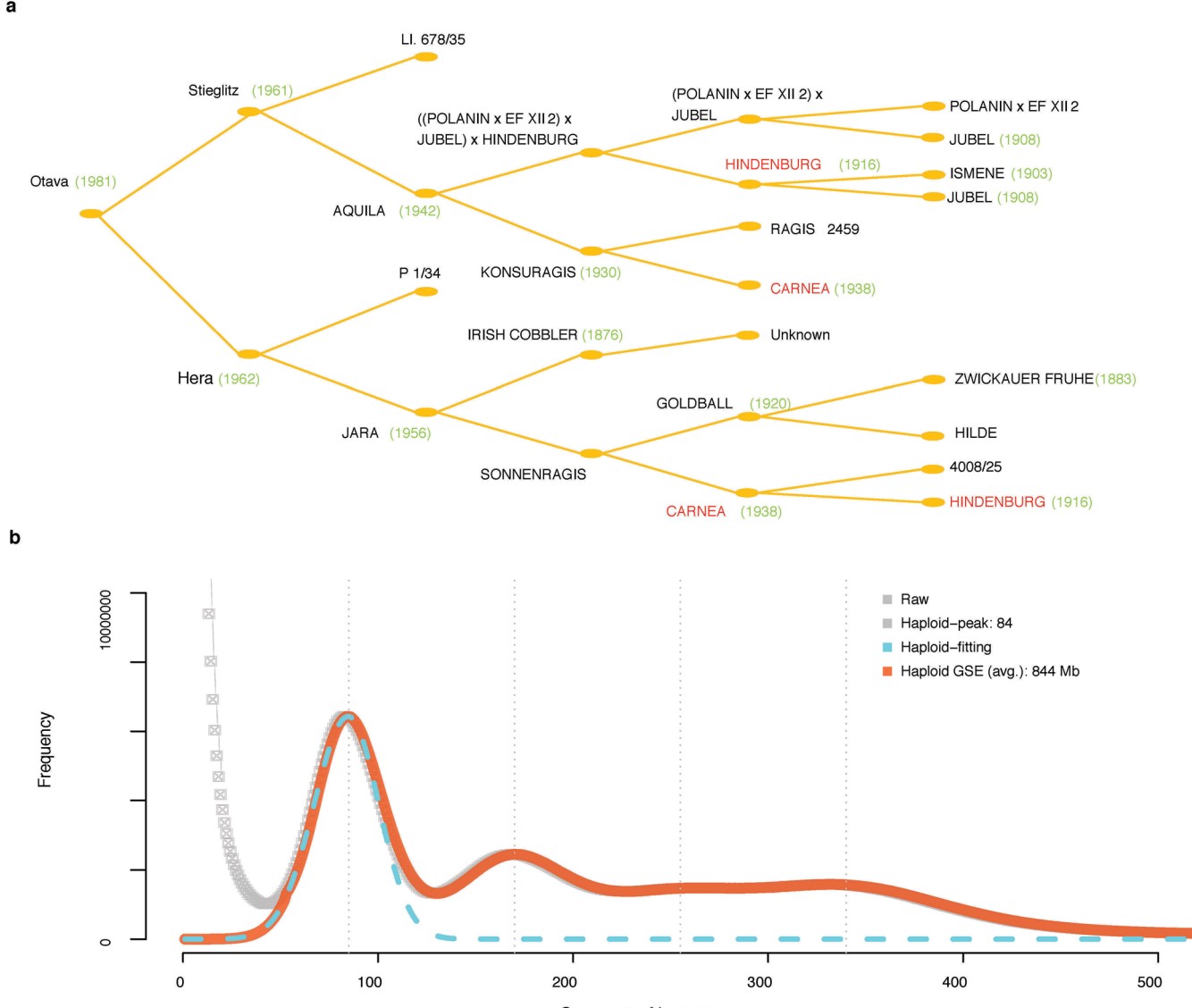

**Extended Data Fig. 1 | General information about the autotetraploid potato cultivar 'Otava'. a**. Pedigree of 'Otava'. The potato cultivar 'Otava' resulted from a cross of 'Hera' and 'Stieglitz' in 1981. Note that 'Hera' and 'Stieglitz' have common ancestors like 'HINDENBURG' (1916) and 'CARNEA' (1938) explaining the presence of IBD regions between the four haplotypes of *Otava* genome. This illustration was modified from the *Potato Pedigree Database* (https://www.plantbreeding.wur.nl/PotatoPedigree). **b**. *K*-mer frequency distribution of the tetraploid genome of 'Otava' and genome size estimation. Note, 844 Mb depicts the estimated haploid genome size, while the tetraploid genome size would be four times the haploid genome size, that is, 3,375 Mb.

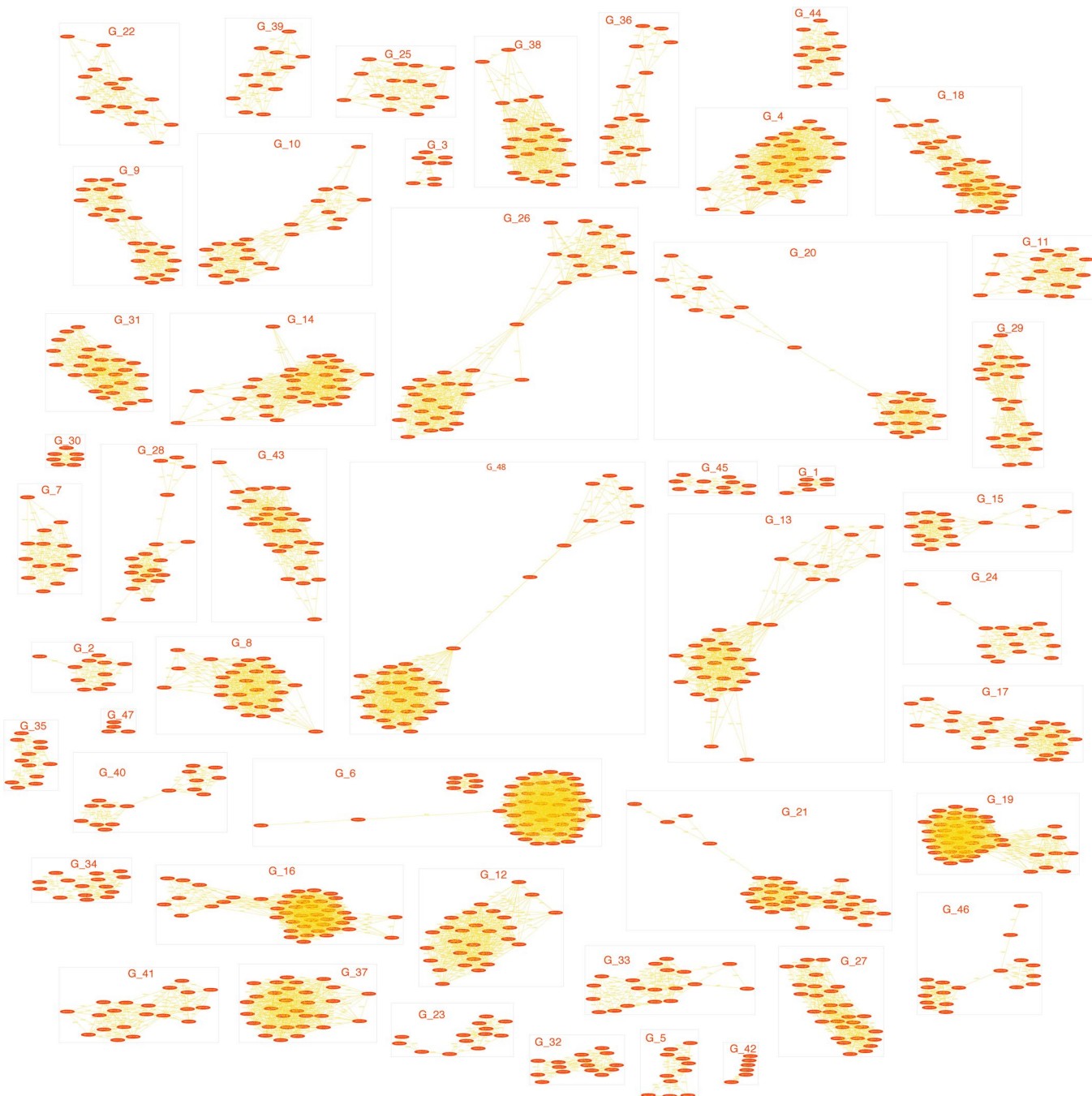

**Extended Data Fig. 2 | Clustering of initially assembled contigs.** The graph (consisting of 48 sub-graphs) visualizes the clustering of haplotigs of over 100 kb into 48 groups representing the 48 haplotype-specific chromosomes. Vertices in red represent contigs, edges in yellow represent a positive correlation (over 0.55) of the PAPs of the connected coverage markers (vertices). All nodes in the same box in gray represent a group of contigs belonging to the same chromosome labeled with G_*i*, where *i* = 1, 2, …, 48. The graph was visualized with *Graphviz* (https://graphviz.org/).

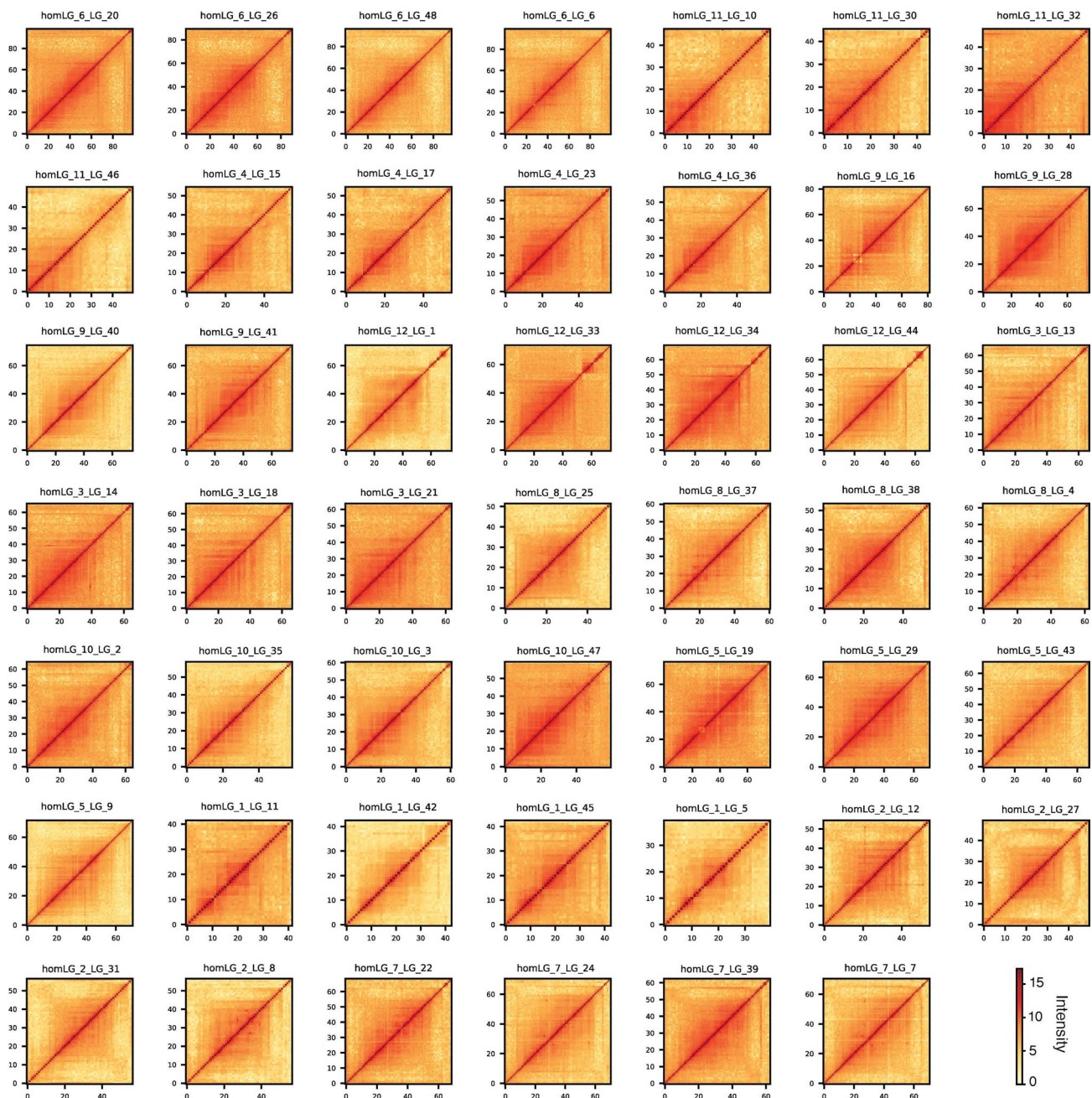

**Extended Data Fig. 3 | Hi-C contact map for each of the 48 haplotype-specific chromosomes.** *HomLG* 6, 11, 4, 9, 12, 3, 8, 10, 5, 1, 2, 7 correspond to LG 1 to 12 of reference genome. For example, *homLG_6* corresponds to LG 1 in the reference assembly (of DM). The four sub-groups (*LG_20*, *LG_26*, *LG_48* and *LG_6*) correspond to the four haplotype-specific chromosomes of 'Otava' (for which the identifiers were defined by the linkage grouping step of gamete binning as given by Extended Data Fig. 2).

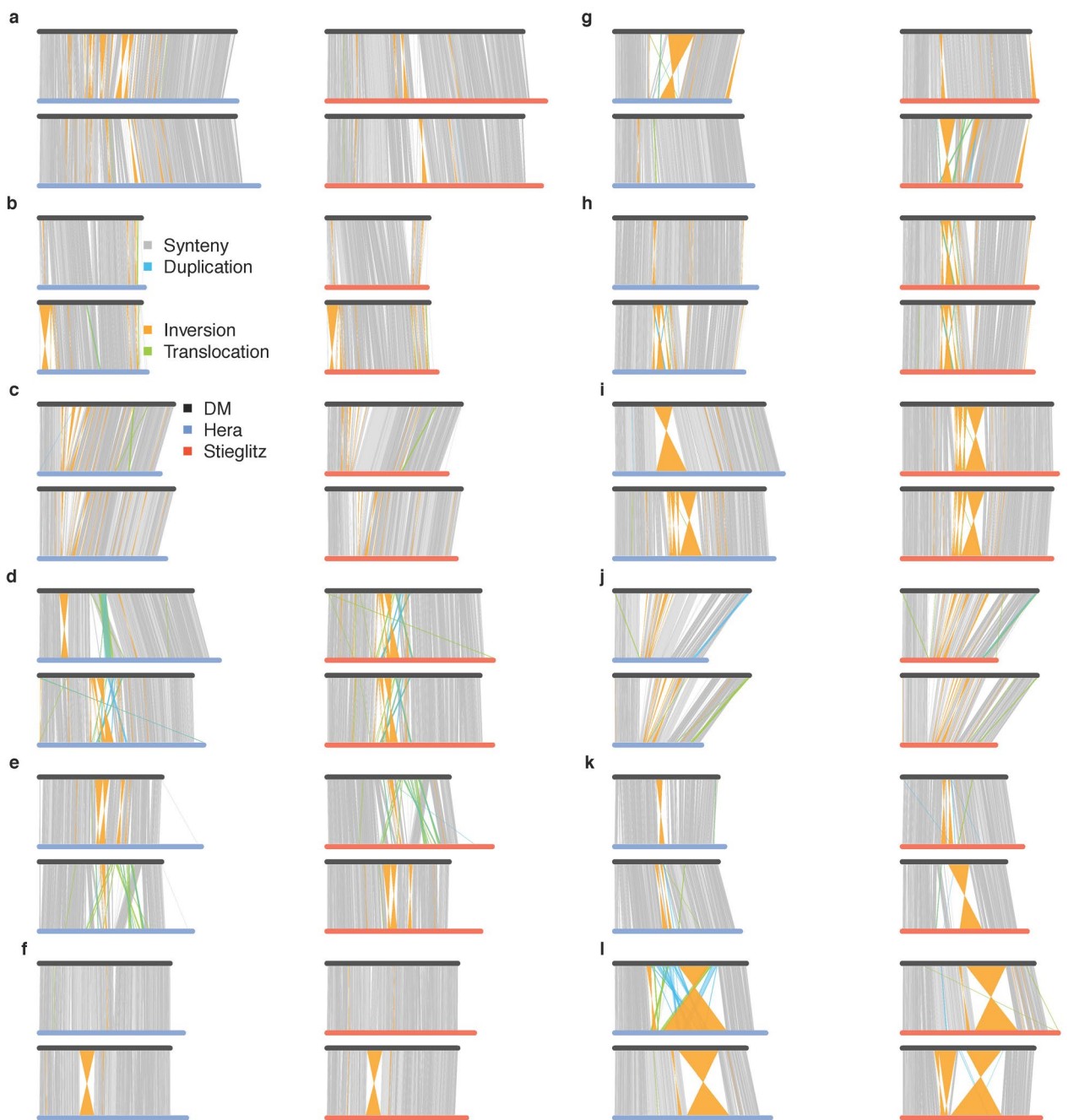

**Extended Data Fig. 4 | Comparison of the 'Otava' assembly to the *DM* assembly within linkage groups 1-12.** a-l: Linkage group (LG) 1-12. Within each LG, there were four haplotype sequences from 'Otava' with two of them inherited from 'Hera' (in light blue) and two inherited from 'Stieglitz' (in light red). Alignments between any two sequences above 50 kb are shown. Specifically, for LG10 (given by **j**), the four haplotype-specific chromosomes showed clear large gaps when aligned to the DM chromosome, indicating that there were potential large chromosomal rearrangements between the two cultivars.

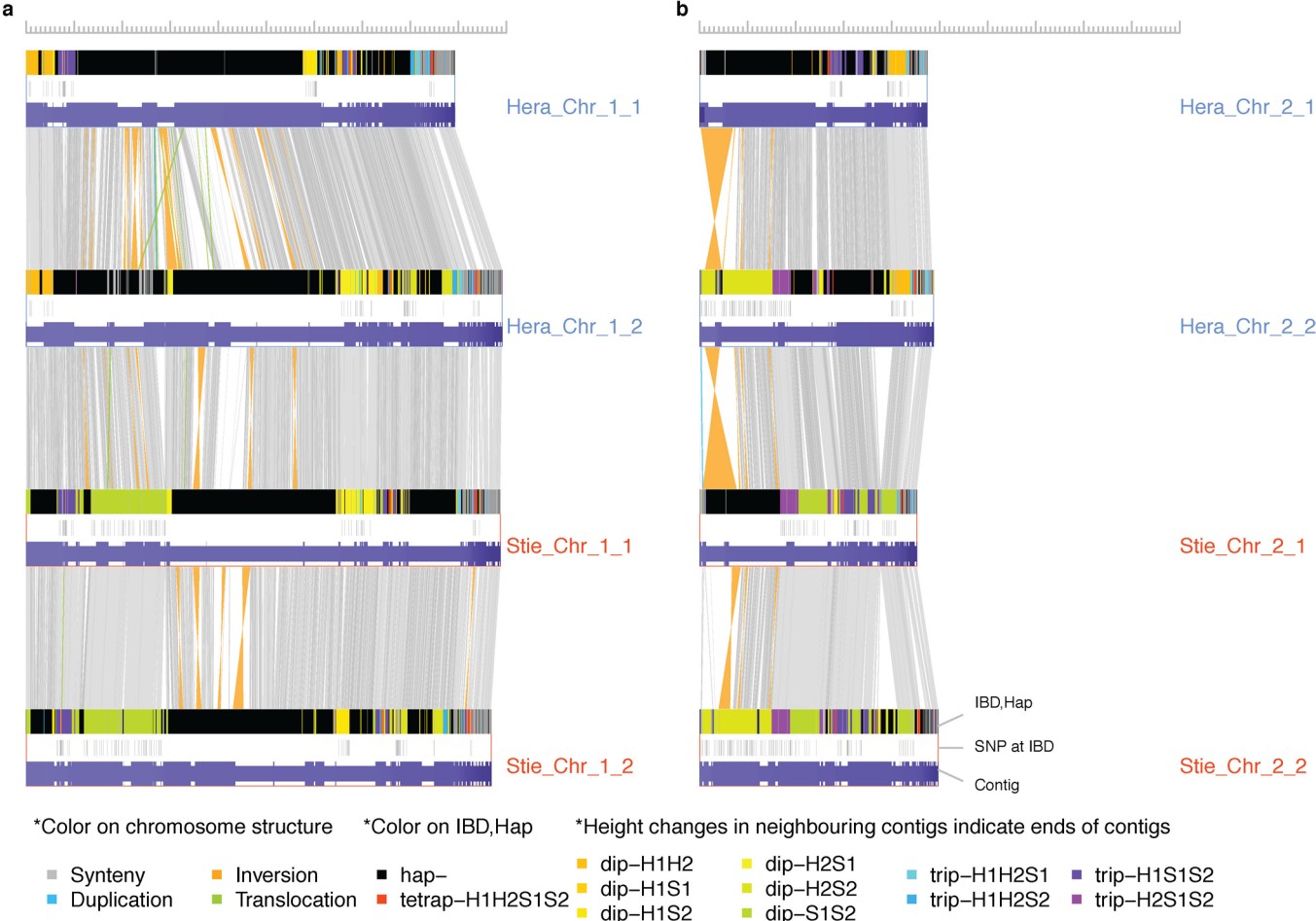

**a**

Hera_Chr_1_1

Hera_Chr_1_2

Stie_Chr_1_1

Stie_Chr_1_2

**b**

Hera_Chr_2_1

Hera_Chr_2_2

Stie_Chr_2_1

IBD,Hap

SNP at IBD

Contig

Stie_Chr_2_2

*Color on chromosome structure          *Color on IBD,Hap          *Height changes in neighbouring contigs indicate ends of contigs

- Synteny          ■ Inversion          ■ hap–
- Duplication          ■ Translocation          ■ tetrap–H1H2S1S2

■ dip–H1H2          ■ dip–H2S1          ■ trip–H1H2S1          ■ trip–H1S1S2
■ dip–H1S1          ■ dip–H2S2          ■ trip–H1H2S2          ■ trip–H2S1S2
■ dip–H1S2          ■ dip–S1S2

**Extended Data Fig. 5 | Comparison of haplotype sequences within linkage groups 1 (a) and 2 (b).** Each of the four horizontal rows represents a haplotype of the chromosome (*Hera* 1, *Hera* 2, *Stieglitz* 1 and *Stieglitz* 2). For each haplotype, three types of information are within each box, top: IBD and unique regions along the chromosome (at 50 kb resolution), middle: distribution of SNPs at IBD regions, and bottom: scaffolded contigs. For each chromosome, three pairwise structural comparisons are shown highlighting inversions (orange), duplications (blue), translocations (green) and syntenic regions (gray). Note that all the regions involving the breakpoints of the SVs ended within contigs. IBD regions can be shared by two (dip), three (trip) or even four (tetrap) haplotypes. The x-axis scale: 0–100 Mb.

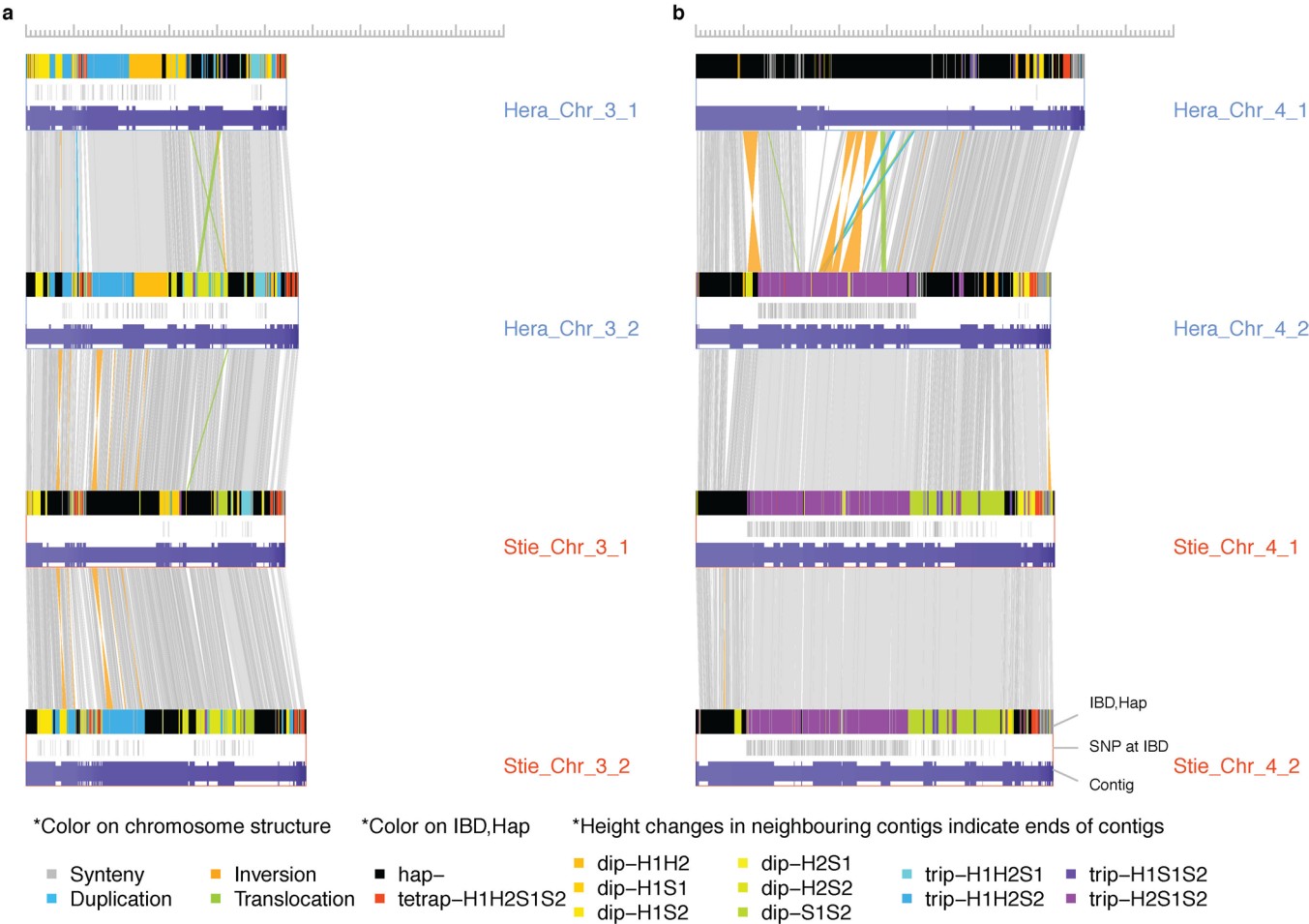

**Extended Data Fig. 6 | Comparison of haplotype sequences within linkage groups 3 (a) and 4 (b).** Each of the four horizontal rows represents a haplotype of the chromosome (*Hera* 1, *Hera* 2, *Stieglitz* 1 and *Stieglitz* 2). For each haplotype, three types of information are within each box, top: IBD and unique regions along the chromosome (at 50 kb resolution), middle: distribution of SNPs at IBD regions, and bottom: scaffolded contigs. For each chromosome, three pairwise structural comparisons are shown highlighting inversions (orange), duplications (blue), translocations (green) and syntenic regions (gray). Note that all the regions involving the breakpoints of the SVs ended within contigs. IBD regions can be shared by two (dip), three (trip) or even four (tetrap) haplotypes. The x-axis scale: 0–100 Mb.

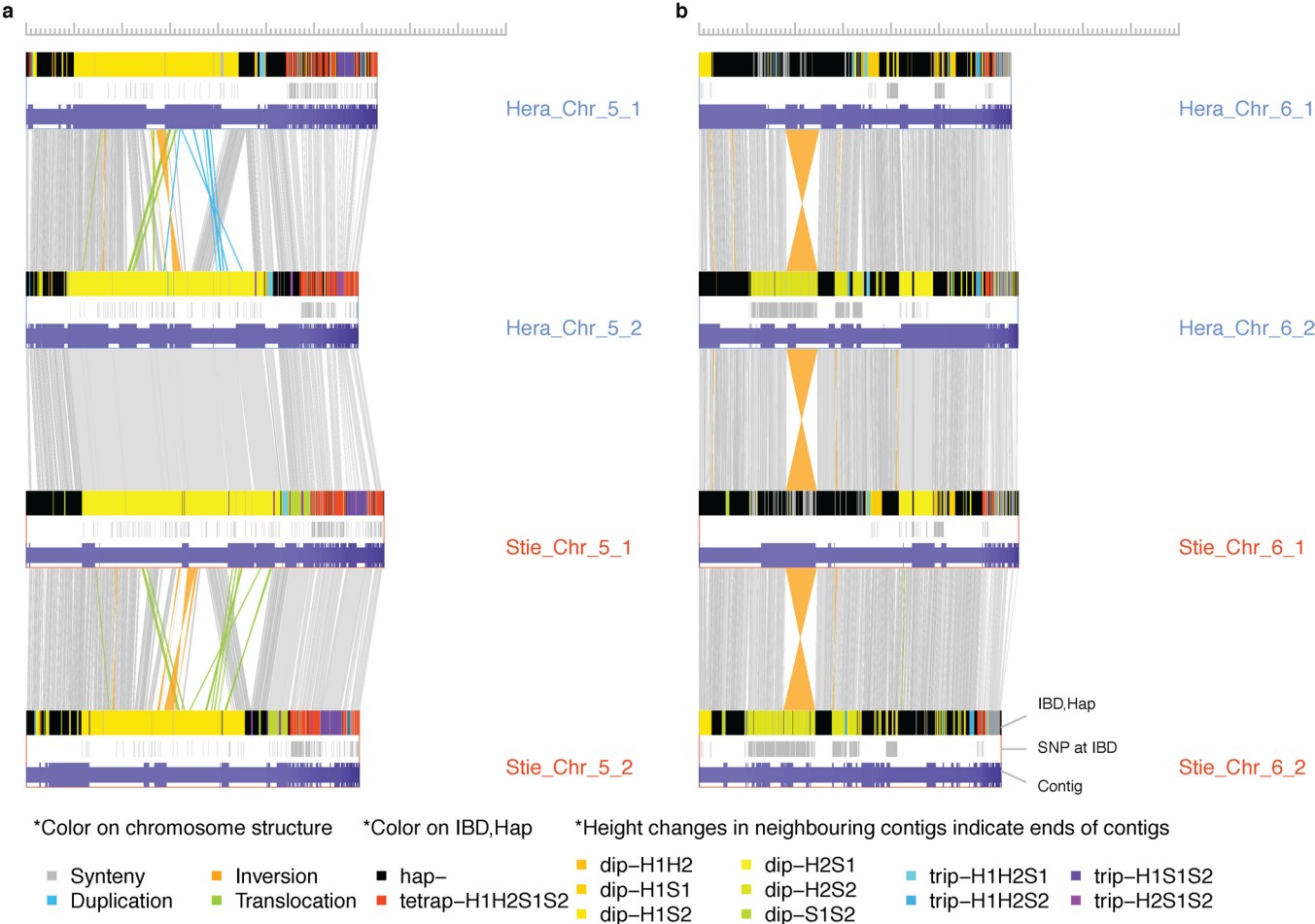

**Extended Data Fig. 7 | Comparison of haplotype sequences within linkage groups 5 (a) and 6 (b).** Each of the four horizontal rows represents a haplotype of the chromosome (*Hera* 1, *Hera* 2, *Stieglitz* 1 and *Stieglitz* 2). For each haplotype, three types of information are within each box, top: IBD and unique regions along the chromosome (at 50 kb resolution), middle: distribution of SNPs at IBD regions, and bottom: scaffolded contigs. For each chromosome, three pairwise structural comparisons are shown highlighting inversions (orange), duplications (blue), translocations (green) and syntenic regions (gray). Note that all the regions involving the breakpoints of the SVs ended within contigs. IBD regions can be shared by two (dip), three (trip) or even four (tetrap) haplotypes. The x-axis scale: 0–100 Mb.

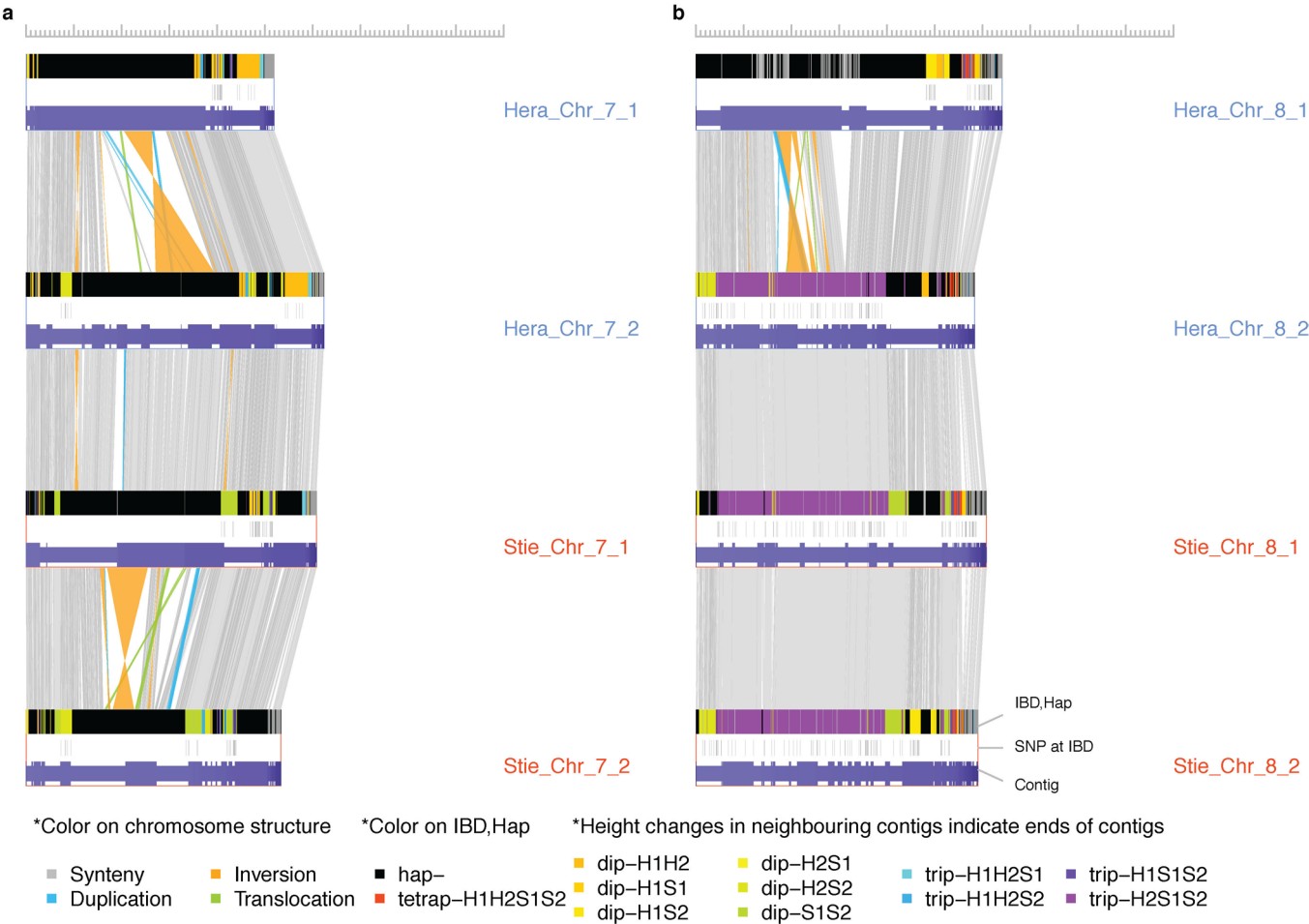

**Extended Data Fig. 8 | Comparison of haplotype sequences within linkage groups 7 (a) and 8 (b).** Each of the four horizontal rows represents a haplotype of the chromosome (*Hera* 1, *Hera* 2, *Stieglitz* 1 and *Stieglitz* 2). For each haplotype, three types of information are within each box, top: IBD and unique regions along the chromosome (at 50 kb resolution), middle: distribution of SNPs at IBD regions, and bottom: scaffolded contigs. For each chromosome, three pairwise structural comparisons are shown highlighting inversions (orange), duplications (blue), translocations (green) and syntenic regions (gray). Note that all the regions involving the breakpoints of the SVs ended within contigs. IBD regions can be shared by two (dip), three (trip) or even four (tetrap) haplotypes. The x-axis scale: 0–100 Mb.

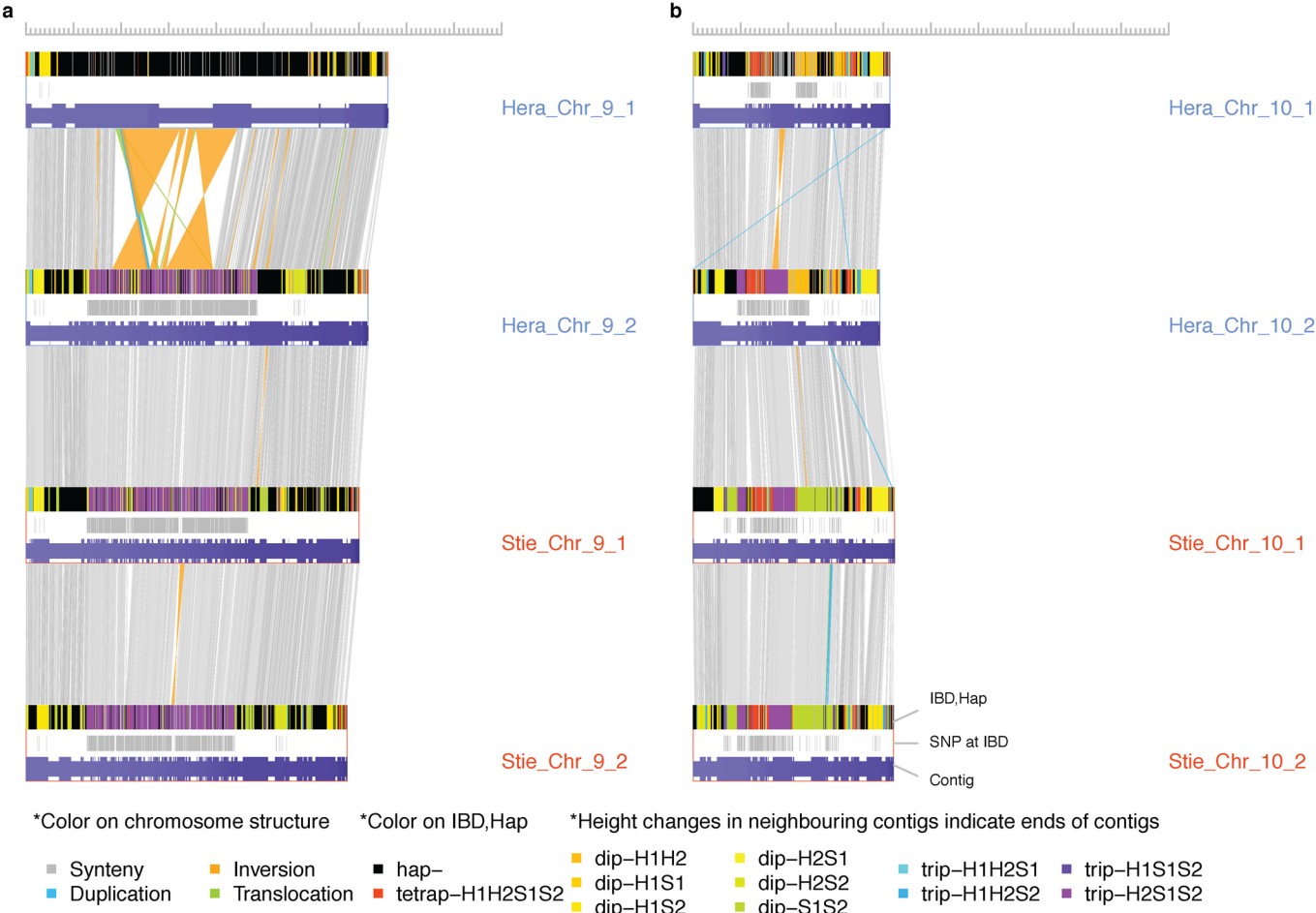

**a**

Hera_Chr_9_1

Hera_Chr_9_2

Stie_Chr_9_1

Stie_Chr_9_2

**b**

Hera_Chr_10_1

Hera_Chr_10_2

Stie_Chr_10_1

IBD,Hap

SNP at IBD

Contig

Stie_Chr_10_2

*Color on chromosome structure  *Color on IBD,Hap  *Height changes in neighbouring contigs indicate ends of contigs

- Synteny
- Duplication
- Inversion
- Translocation
- hap–
- tetrap–H1H2S1S2
- dip–H1H2
- dip–H1S1
- dip–H1S2
- dip–H2S1
- dip–H2S2
- dip–S1S2
- trip–H1H2S1
- trip–H1H2S2
- trip–H1S1S2
- trip–H2S1S2

**Extended Data Fig. 9 | Comparison of haplotype sequences within linkage groups 9 (a) and 10 (b).** Each of the four horizontal rows represents a haplotype of the chromosome (*Hera* 1, *Hera* 2, *Stieglitz* 1 and *Stieglitz* 2). For each haplotype, three types of information are within each box, top: IBD and unique regions along the chromosome (at 50 kb resolution), middle: distribution of SNPs at IBD regions, and bottom: scaffolded contigs. For each chromosome, three pairwise structural comparisons are shown highlighting inversions (orange), duplications (blue), translocations (green) and syntenic regions (gray). Note that all the regions involving the breakpoints of the SVs ended within contigs. IBD regions can be shared by two (dip), three (trip) or even four (tetrap) haplotypes. The x-axis scale: 0–100 Mb.

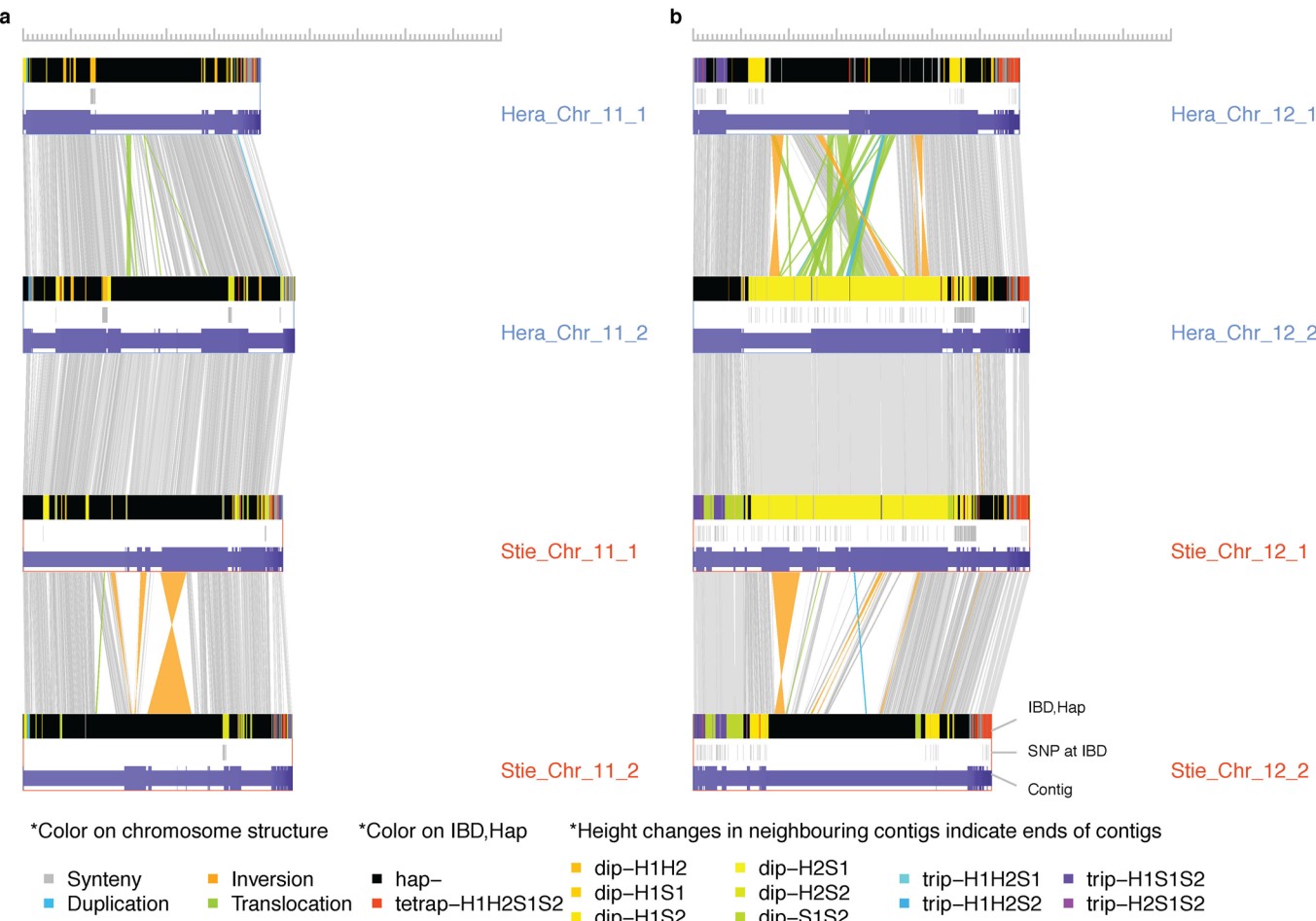

a

Hera_Chr_11_1

Hera_Chr_11_2

Stie_Chr_11_1

Stie_Chr_11_2

b

Hera_Chr_12_1

Hera_Chr_12_2

Stie_Chr_12_1

IBD,Hap

SNP at IBD

Contig

Stie_Chr_12_2

*Color on chromosome structure *Color on IBD,Hap *Height changes in neighbouring contigs indicate ends of contigs

| | | | | | | |
|---|---|---|---|---|---|---|
| ▪ Synteny | ▪ Inversion | ▪ hap– | ▪ dip–H1H2 | ▪ dip–H2S1 | ▪ trip–H1H2S1 | ▪ trip–H1S1S2 |
| ▪ Duplication | ▪ Translocation | ▪ tetrap–H1H2S1S2 | ▪ dip–H1S1 | ▪ dip–H2S2 | ▪ trip–H1H2S2 | ▪ trip–H2S1S2 |
| | | | ▪ dip–H1S2 | ▪ dip–S1S2 | | |

**Extended Data Fig. 10 | Comparison of haplotype sequences within linkage groups 11 (a) and 12 (b).** Each of the four horizontal rows represents a haplotype of the chromosome (*Hera* 1, *Hera* 2, *Stieglitz* 1 and *Stieglitz* 2). For each haplotype, three types of information are within each box, top: IBD and unique regions along the chromosome (at 50 kb resolution), middle: distribution of SNPs at IBD regions, and bottom: scaffolded contigs. For each chromosome, three pairwise structural comparisons are shown highlighting inversions (orange), duplications (blue), translocations (green) and syntenic regions (gray). Note that all the regions involving the breakpoints of the SVs ended within contigs. IBD regions can be shared by two (dip), three (trip) or even four (tetrap) haplotypes. The x-axis scale: 0–100 Mb.

# Reporting Summary

## Statistics

For all statistical analyses, confirm that the following items are present in the figure legend, table legend, main text, or Methods section.

| n/a | Confirmed | |
|---|---|---|
| ☐ | ☒ | The exact sample size (*n*) for each experimental group/condition, given as a discrete number and unit of measurement |
| ☒ | ☐ | A statement on whether measurements were taken from distinct samples or whether the same sample was measured repeatedly |
| ☐ | ☒ | The statistical test(s) used AND whether they are one- or two-sided *Only common tests should be described solely by name; describe more complex techniques in the Methods section.* |
| ☒ | ☐ | A description of all covariates tested |
| ☐ | ☒ | A description of any assumptions or corrections, such as tests of normality and adjustment for multiple comparisons |
| ☐ | ☒ | A full description of the statistical parameters including central tendency (e.g. means) or other basic estimates (e.g. regression coefficient) AND variation (e.g. standard deviation) or associated estimates of uncertainty (e.g. confidence intervals) |
| ☐ | ☒ | For null hypothesis testing, the test statistic (e.g. *F*, *t*, *r*) with confidence intervals, effect sizes, degrees of freedom and *P* value noted *Give P values as exact values whenever suitable.* |
| ☒ | ☐ | For Bayesian analysis, information on the choice of priors and Markov chain Monte Carlo settings |
| ☒ | ☐ | For hierarchical and complex designs, identification of the appropriate level for tests and full reporting of outcomes |
| ☒ | ☐ | Estimates of effect sizes (e.g. Cohen's *d*, Pearson's *r*), indicating how they were calculated |

*Our web collection on statistics for biologists contains articles on many of the points above.*

## Software and code

Policy information about availability of computer code

| Data collection | No code used for data collection. |
|---|---|
| Data analysis | All routine analysis were performed using tools publicly available. Barcodes were corrected using cellranger (10x Genomics, v1.1.0). Read quality control: Trimmomatic (v0.39). Short/long reads were aligned using bowtie2 (2.2.8)/minimap2 (2.20-r1061), hisat2 (v2.2.0). BAM, VCF file processing and sequencing depth analysis were performed using samtools (v1.9) and bedtools (v2.27.1). PacBio sequence reads were assembled using hifiasm (v0.7). Contig polishing: pilon (v1.22) and racon (1.4.10). Structural variations were identified using SyRI (v1.0) based on minimap2 genome alignments. Methylation calling was performed with bismark (v0.23.0) pipeline. Genome size was estimated using findGSE (v1.0). k-mers were counted and handled using jellyfish (v2.2.10), kmc3 (v3.1.1) and Merqury (v1.3). Statistical analysing including t-test and correlation test were performed using R (v3.5.1). Blast of nucleotides was performed with blast tool kit (v2.10.0+). Hi-C data was processed using ALLHiC (v0.9.8) and SALSA2 (v2.2) pipeline. BUSCO analysis using BUSCO (version 4.1.4). Genome annotation: Augustus (version 3.2.3), GlimmerHMM (version 3.0.1), SNAP (v2006-07-28), exonerate (v2.2.0), StringTie (v1.3.4d), EvidenceModeler (version not available), RepeatModeler (v2.0.1), RepeatMasker (open-4.0), Infernal (v1.1). RNA-seq quantification: HTSeq (v0.13.5). Replicates of expression and methylation clustering: package hclust (v3.5.1) in R. Protein sequence alignment: blastp (v2.10.0+) and Scipio (v1.4.1). Functional annotation: InterProscan (v5.48) and R package ClusterProfiler (v3.14). |

Protein clustering: OrthoFinder (v2.2.6).
Transcript checking: FEELnc (v.0.1.1) and CPC2 (v0.9-r2).
Custome code and scripts supporting this work is available at github.com/schneeberger-lab/GameteBinning_tetraploid or Zenodo under Creative Commons Attribution 4.0 International license (DOI: 10.5281/zenodo.5775114).

For manuscripts utilizing custom algorithms or software that are central to the research but not yet described in published literature, software must be made available to editors and reviewers. We strongly encourage code deposition in a community repository (e.g. GitHub). See the Nature Portfolio guidelines for submitting code & software for further information.

## Data

Policy information about availability of data

All manuscripts must include a data availability statement. This statement should provide the following information, where applicable:
- Accession codes, unique identifiers, or web links for publicly available datasets
- A description of any restrictions on data availability
- For clinical datasets or third party data, please ensure that the statement adheres to our policy

High-throughput sequencing data analyzed in this project are available under NCBI BioProject: PRJNA751899. This Whole Genome Shotgun project (including assembly and annotation) has been deposited at DDBJ/ENA/GenBank under the accessions JAIVGA000000000, JAIVGB000000000, JAIVGC000000000 and JAIVGD000000000. The version described in this paper is version JAIVGA010000000, JAIVGB010000000, JAIVGC010000000 and JAIVGD010000000. The genome assembly and gene annotation of 'Otava' are also available on Spud DB (http://spuddb.uga.edu/)6. Source data are provided at Zenodo under Creative Commons Attribution 4.0 International license (DOI: 10.5281/zenodo.5796752). Genome information of DM and RH used in this study are available on Spud DB (http://spuddb.uga.edu/).

All the databases/datasets used in the study are along with appropriately accessible links/accession-codes in the manuscript under the "Data availability" section as well as in this reporting summary.

# Field-specific reporting

Please select the one below that is the best fit for your research. If you are not sure, read the appropriate sections before making your selection.

☒ Life sciences　　　☐ Behavioural & social sciences　　　☐ Ecological, evolutionary & environmental sciences

For a reference copy of the document with all sections, see nature.com/documents/nr-reporting-summary-flat.pdf

# Life sciences study design

All studies must disclose on these points even when the disclosure is negative.

| | |
|---|---|
| Sample size | Three samples/cultivars were selected, including Hera, Stieglitz and Otava, which make a trio for genome and haplotyping analysis. |
| Data exclusions | No data were excluded from the analysis. |
| Replication | Three biological replicates were respectively used for both RNA and methylation sequencing and analysis. All attempts at replication were successful. |
| Randomization | This is not relevant to this study, as it is about assembly and analysis of a single cultivar. |
| Blinding | No group allocation was needed in this study. |

# Reporting for specific materials, systems and methods

We require information from authors about some types of materials, experimental systems and methods used in many studies. Here, indicate whether each material, system or method listed is relevant to your study. If you are not sure if a list item applies to your research, read the appropriate section before selecting a response.

## Materials & experimental systems

| n/a | Involved in the study |
|---|---|
| ☒ | ☐ Antibodies |
| ☒ | ☐ Eukaryotic cell lines |
| ☒ | ☐ Palaeontology and archaeology |
| ☒ | ☐ Animals and other organisms |
| ☒ | ☐ Human research participants |
| ☒ | ☐ Clinical data |
| ☒ | ☐ Dual use research of concern |

## Methods

| n/a | Involved in the study |
|---|---|
| ☒ | ☐ ChIP-seq |
| ☒ | ☐ Flow cytometry |
| ☒ | ☐ MRI-based neuroimaging |

