## [Peer Review File · Nature Genetics]

Peer Review Information

Manuscript Title: Chromosome-scale and haplotype-resolved genome assembly of a tetraploid potato cultivar

Corresponding author name(s): Professor Korbinian Schneeberger

Reviewer Comments & Decisions:

Decision Letter, initial version:
--

30th Jun 2021

Dear Professor Schneeberger,

Your Letter, "Chromosome-scale and haplotype-resolved genome assembly of a tetraploid potato cultivar" has now been seen by 3 referees. You will see from their comments below that while they find your work of interest, some important points are raised. We are interested in the possibility of publishing your study in Nature Genetics, but would like to consider your response to these concerns in the form of a revised manuscript before we make a final decision on publication.

As you will see from these comments, all referees have identified aspects of the analyses and the presentation that need to be improved. In particular, reviewers #1 and #2 have serious concerns regarding the degree of biological insights and the potential impact. Please extend the work taking into account some of reviewer suggestions to enhance its scientific impact and novelty. Please also make the data available for evaluation as noted by reviewer #2. Please address all referees' points as thoroughly as possible.

We therefore invite you to revise your manuscript taking into account all reviewer and editor comments. Please highlight all changes in the manuscript text file. At this stage we will need you to upload a copy of the manuscript in MS Word .docx or similar editable format.

*2) If you have not done so already please begin to revise your manuscript so that it conforms to our Letter format instructions, available [here](http://www.nature.com/ng/authors/article_types/index.html). Refer also to any guidelines provided in this letter.

[REDACTED]

We hope to receive your revised manuscript within three to six months. If you cannot send it within this time, please let us know.

Sincerely,

Wei

Wei Li, PhD
Senior Editor
Nature Genetics
One New York Plaza, 47th Fl.
New York, NY 10004, USA
www.nature.com/ng

Reviewers' Comments:

Reviewer #1:

Remarks to the Author:

This manuscript shows a very impressive haplotype-resolved analysis of a tetraploid potato. The authors used a combination of the relatively recently developed PacBio HiFi sequencing with gamete binning using Illumina sequencing to create a high-resolution assembly of all four copies of the genome of this tetraploid.

This represents a significant technical advance, as the resolution of tetraplotigs as described here has, as far as I know, not been described before. The methods for gamete binning using pollen are relatively straightforward and could be widely applied to other polyploid crops. There are also some implications for plant breeding, in terms of being able to determine identity by descent at base-pair resolution. Such methods will be critical for understanding the origin and propagation of heterosis in polyploid crops, among other applications. The authors correctly point out that the diversity in diploid potato varieties could be greatly enhanced by further application of these types of assembly techniques, although the cost is likely prohibitive currently for most breeding applications.

As far as scientific discovery is concerned, while important, the science is of lower impact than the methods here. Diploid potato genomes (both doubled haploid and haplotype-resolved) are available already, and the extensive identity by descent makes this genome sequence of limited utility as a reference for breeding purposes. The relevance of the work presented here to breeding is therefore somewhat oversold, though widespread application of these methods across tetraploid potato would definitely have a significant impact on approaches to breeding.

The quality of the assembly appears to be very high, and all of the methods are appropriate. In terms of the reported results this appears to be very strong work indeed.

Functional genomics and characterization of RNA and small RNA / lncRNA would be a nice addition. The impact of the high levels of identity between homeologous regions on methylation would also be good to know. Analysis of the degree to which different transposon families contribute to the observed structural variation would be useful also. Characterization of the gene expression consequences of structural polymorphism and haplotype divergence would increase the potential for impactful scientific discovery.

Reviewer #2:

Remarks to the Author:

I have read the manuscript by Sun et al. entitled "Chromosome-scale and haplotype-resolved genome assembly of a tetraploid potato cultivar" with great interest. The authors describe a strategy based on several sequencing technologies that lead to a high-quality assembly of a tetraploid genome. Indeed, the results obtained using the gamete binning method are impressive and allow correct resolution of haplotypes. I congratulate the authors for obtaining an assembly of this quality.

Given the existence of multiple other haploid or diploid potato genomes (1,2,3), the new genome provides only an incremental increase in our knowledge of the potato genomes. Likewise, the sequencing and assembly strategy has already been published by the authors (4). The authors compare the four haplotypes, but the analysis presented here are mainly descriptive. Genome assemblies of this quality do represent a valuable resource for the scientific community, but in my opinion the novel scientific insights described in this manuscript are limited.

1- <https://www.g3journal.org/content/10/10/3489>

2- <https://academic.oup.com/gigascience/article/9/9/giaa100/5910251>

3- <https://www.nature.com/articles/s41588-020-0699-x>

4- <https://genomebiology.biomedcentral.com/articles/10.1186/s13059-020-02235-5>

I am embarrassed by the fact that the data (raw reads, tetraploid assembly, github repository) are not available. Therefore, I could not verify the quality of the scientific work even if the article seems to be of high quality. Authors must share the data with the reviewers.

I have a few more specific points/questions below.

(1) Figure 1a is oversimplified and I think the authors should try to combine it with Figure S5 which is more precise as it describes collapse contigs.

(2) L106-107 : "Reads aligned to diplotigs, triplotigs and tetraplotigs were randomly assigned to one of the respective haplotypes." In this case, even if the assembler had collapsed these regions, they may contain small variations between the haplotypes, and this random assignment can lead to mixing the haplotypes.

(3) L115-116 : the authors should explain in more detail why the LG10 is shorter in their assembly compared to existing assemblies.

(4) The assembly has been polished with Pilon which performs an error correction by examining the pileup of bases from the reads. Unfortunately this strategy can lead to haplotype mixing, did the authors check the local accuracy ?

(5) On Figure2, the light and dark blue boxes are not easily identifiable.

(6) The authors report that a large fraction of the genes were not present in the four haplotypes. Could this result be an artefact of the annotation process? I read that the authors use the "--percent 60" option of exonerate but how Hisat and StringTie handled the presence of four haplotypes? More generally, I was a little surprised by this result, because the BUSCO scores are high in each of the four

haplotypes. Does this mean that BUSCO gene sets are biased ?

Reviewer #3:

Remarks to the Author:

Review of the paper entitled "Chromosome-scale and haplotype-resolved genome assembly of a tetraploid potato cultivar" by Sun, H. et al. submitted to Nature Genetics.

This paper reports de novo assembly of four haplotypes of a tetraploid potato (*Solanum tuberosum*) cultivar. Potato is the most important root crop of the world. However, its autotetraploid genome and high levels of inbreeding depression hampered application of modern breeding techniques to the crop. Potato genome sequences have been reported previously using a di-haploid individual derived from a homozygous diploid potato (The Potato Genome Sequencing Consortium, 2011) and a heterozygous diploid potato (Zhou et al. 2020). In the latter study, haplotype-resolved genome assembly was reported using linkage information obtained by WGS of selfed F2 progeny.

The submitted paper is the first report of haplotype-resolved genome assembly of a cultivar Otava with a tetraploid genome, which comprises the majority of the cultivated potato. First, the authors obtained PacBio long reads and contigs. To obtain the four haplotypes, the authors employed Illumina WGS of gametic genomes of 717 pollens (following the approach of Shi et al. 2019) from the individual Otava. Gametic reads were mapped to the contigs, and presence/absence of reads in the contigs allowed grouping of the contigs to the four haplotypes. De novo assembly was conducted again for the reads belonging to each of the four groups, and the generated haplotypes were validated by Hi-C. Also, genome sequences of Stieglitz and Hera, the two parents of Otava, was sequenced to confirm the haplotypes.

Comparison of the four haplotypes revealed that 50% of the genomic regions were identical (IBD) in at least two haplotypes. On the other hand, higher divergence between the haplotypes were marked by large-scale structural changes involving inversion, duplication, translocation, suggesting introgression from wild species. Gene comparison among the four haplotypes showed that only 53.6% of the genes were present in all four haplotypes, and the rest were share by a subset of haplotypes. Remarkably, the genes with four identical alleles show enrichment in GO of photosynthesis, chlorophyll binding and translation, suggesting selective optimization of the tetraploid potato.

By employing an innovative and solid genomics approach, this paper revealed genome structure of autopolyploid crop species for the first time. This study will be a benchmark study of genomics of polyploid species which are common in crops.

Specific points:

1/ Figure S5, S6, S7, Figure 3b: Allocation of pollen reads to contigs to group them into four haplotypes for 12 chromosomes is overall clear. However, it is not clear how the authors solved the junction between the haplotype-unique regions and the shared regions. As an example, in Fig. S15, Hera_Chr_4_2 (Hera_2), Stie_Chr_4_2 (Stie_2), Stie_Chr_4_1 (Stie_1) share a ~30Mb trip-H1S1S2 IBD region (IBD) in the middle between the unique regions on the left (left) and right (right) sides of the chromosome. With the method described, it seems possible that "Hera_2 left - IBD - Stie_1 right" or "Stie_1 left - IBD - Stie_2 right" configurations are also obtained. An explanation will benefit the understanding of readers.

2/ The paper does not mention recombination of haplotypes in the gametes. A description how it affects and not affect the method is needed.

Author Rebuttal to Initial comments

Reviewers' Comments:

Reviewer #1:

Remarks to the Author:

This manuscript shows a very impressive haplotype-resolved analysis of a tetraploid potato. The authors used a combination of the relatively recently developed PacBio HiFi sequencing with gamete binning using Illumina sequencing to create a high-resolution assembly of all four copies of the genome of this tetraploid.

This represents a significant technical advance, as the resolution of tetraplotigs as described here has, as far as I know, not been described before. The methods for gamete binning using pollen are relatively straightforward and could be widely applied to other polyploid crops. There are also some implications for plant breeding, in terms of being able to determine identity by descent at base-pair resolution. Such methods will be critical for understanding the origin and propagation of heterosis in polyploid crops, among other applications. The authors correctly point out that the diversity in diploid potato varieties could be greatly enhanced by further application of these types of assembly techniques, although the cost is likely prohibitive currently for most breeding applications.

As far as scientific discovery is concerned, while important, the science is of lower impact than the methods here. Diploid potato genomes (both doubled haploid and haplotype-resolved) are available already, and the extensive identity by descent makes this genome sequence of limited utility as a reference for breeding purposes. The relevance of the work presented here to breeding is therefore somewhat oversold, though widespread application of these methods across tetraploid potato would definitely have a significant impact on approaches to breeding.

The quality of the assembly appears to be very high, and all of the methods are appropriate. In terms of the reported results this appears to be very strong work indeed.

Thank you very much for highlighting the quality and advancements of our work. We agree that sequencing costs are still high in particular when many genomes are targeted. However, considering the decline of costs of other genomic technologies in the past, we believe that even the assembly of multiple potato genomes will become realistic soon. In this respect, we also agree with the reviewer that the biggest impact of our work on breeding might be the assembly method itself. While the four assembled haplotypes give insights into the tetraploid nature of potato and help to reduce many of the current reference-bias problems, soon there will be more and more genomes available. Our work shows how such tetraploid genomes can be assembled and what to expect from them.

Comment 1.1 Functional genomics and characterization of RNA and small RNA / lncRNA would be a nice addition.

***Response 1.1:** To complete the functional characterization of the genome, we have now analyzed noncoding RNAs, including tRNA, rRNA, snRNA, snoRNA, miRNA and long noncoding RNA (lncRNA) in all four haplotype assemblies respectively. In general, the four haplotypes carried comparable amounts of small RNA and lncRNA, which accounted for 33.9 Mb across the entire genome. A description on this has been added to the last paragraph of Section "Genome assembly of a tetraploid potato" (Supplementary Information Data S4).*

Comment 1.2 The impact of the high levels of identity between homeologous regions on methylation would also be good to know.

***Response 1.2:** To address this, we have now generated three replicates of the DNA methylome from the leaf tissue of 'Otava' (Supplementary Information: sections of "Enzymatic methylome library preparation and sequencing ('Otava')" and "Methylation analysis"; Fig. S28; Data S1). We found that the IBD blocks carried slightly but significantly higher methylation levels as compared to the homologous, but non-shared regions in the other haplotypes (line 266-268; Supplementary Information Fig. S27). This result might be somewhat unexpected as IBD blocks can change within one generation, while DNA methylation can remain stable over generations. In future studies, it will be interesting to analyze how methylated regions regulate haplotype-specific expression, how they segregate in the cultivated potatoes, and how this is affected by the ancestry of the individual haplotypes.*

Comment 1.3 Analysis of the degree to which different transposon families contribute to the observed structural variation would be useful also.

***Response 1.3:** In response to this, we have analyzed transposon element families within and close to structural variations. We found that Gypsy and Copia retrotransposons superfamilies were strikingly enriched in duplications and translocations, but not in inversions (Supplementary Information Fig. S24; Supplementary Information Data S8), which is reminiscent of other plant genomes, where Gypsies and Copias were the most active transposable element families (line 195-198).*

Comment 1.4 Characterization of the gene expression consequences of structural polymorphism and haplotype divergence would increase the potential for impactful scientific discovery.

Response 1.4: Thank you for this suggestion, which triggered us to analyze gene expression within the tetraploid genome in general, and more specifically to analyze allele-specific gene expression. To do this, we generated RNA-seq data from the 'Otava' leaf transcriptome in three replicates (Supplementary Information: sections of "RNA library preparation and sequencing ('Otava')" and "Allelic expression analysis"; Fig. S28; Data S1). At the genome-wide level, the expression levels of all four haplotypes were highly similar suggesting the absence of a sub-genome dominance in this tetraploid genome (Fig. 4d; Supplementary Information Fig. S26). However, consistent with increased expression levels of duplicated genes, we found that genes with a higher number of allelic copies were generally higher expressed than those genes with a lower number of allelic copies (Fig. 4e). Although haplotype-wide gene expression was generally similar between the haplotypes, we found that ~10% of the genes featured differentially expressed alleles (Fig. 4f). Among those, 25% genes showed strong correlations between the allele-specific expression and methylation patterns, suggesting that DNA methylation is one important component in the regulation of allele-specific expression in tetraploid potato (line 254-272).

Reviewer #2:

Remarks to the Author:

I have read the manuscript by Sun et al. entitled "Chromosome-scale and haplotype-resolved genome assembly of a tetraploid potato cultivar" with great interest. The authors describe a strategy based on several sequencing technologies that lead to a high-quality assembly of a tetraploid genome. Indeed, the results obtained using the gamete binning method are impressive and allow correct resolution of haplotypes. I congratulate the authors for obtaining an assembly of this quality.

Thank you very much for highlighting the high quality of our work.

Comment 2.1 Given the existence of multiple other haploid or diploid potato genomes (1,2,3), the new genome provides only an incremental increase in our knowledge of the potato genomes. Likewise, the sequencing and assembly strategy has already been published by the authors (4). The authors compare the four haplotypes, but the analysis presented here are mainly descriptive. Genome assemblies of this quality do represent a valuable resource for the scientific community, but in my opinion the novel scientific insights described in this manuscript are limited.

1-<https://www.g3journal.org/content/10/10/3489>

2-<https://academic.oup.com/gigascience/article/9/9/giaa100/5910251>

3-<https://www.nature.com/articles/s41588-020-0699-x>

4-<https://genomebiology.biomedcentral.com/articles/10.1186/s13059-020-02235-5>

Response 2.1: We respectfully disagree with the reviewer. While there are some haplotype assemblies published, none of them was assembled in the context of a tetraploid genome. In fact, we would argue that the analysis of a diploid potato genome (even when derived from a tetraploid) does not allow for the full understanding of the tetraploid genome (e.g. like the degree of IBD). Moreover, any functional analysis (like our newly added analysis on allele-specific expression) heavily relies on the assembly of the actual, tetraploid genome, which can now be performed with our gamete binning method.

Although the gamete binning method has been published before, we had to improve it to polyploid genomes to face significantly greater challenges as compared to an application to diploids. The key difference is the introduction of what we called "coverage markers" for linkage grouping, which allowed us to genotype individual haplotypes even in presence of multiple, highly divergent haplotypes. We are convinced that this method will have a strong impact on the analysis of other potato genomes as well as on other polyploid crops with similar genome complexity.

In the updated manuscript, we have now generated and analyzed RNA-seq and genome-wide DNA methylation data to get first insights into allele-specific gene expression and its regulation in this tetraploid genome and thereby strengthened the discoveries described in the earlier version of the manuscript.

Comment 2.2 I am embarrassed by the fact that the data (raw reads, tetraploid assembly, github repository) are not available. Therefore, I could not verify the quality of the scientific work even if the article seems to be of high quality. Authors must share the data with the reviewers.

Response 2.2: We do apologize that the data was not uploaded to any public database in time. Our assembly was the first fully assembled tetraploid genomes (i.e. without a primary haplotype) uploaded to NCBI, which required the establishment of new strategies regarding the haplotype genome organization at NCBI and took much more time than anticipated.

The four haplotype genome assemblies and annotations have now been deposited at DDBJ/ENA/GenBank under the accession numbers JAIVGA000000000, JAIVGB000000000, JAIVGC000000000 and JAIVGD000000000. All sequencing data have been uploaded to the NCBI BioProject PRJNA751899 (please note, that some of the short read datasets are still under screening at NCBI but are expected to be online in a few days - please see the attached email from NCBI confirming that they will release the data soon).

In addition, with a kind support from Robin Buell (University of Georgia) the genome assembly and annotation were also integrated into the community-wide known Spud DB webpage (Potato Genomics Resource: http://spuddb.uga.edu/otava_potato_download.shtml). Code, scripts and pipelines used in this work can be downloaded from http://github.com/schneebergerlab/GameteBinning_tetraploid.

I have a few more specific points/questions below.

Comment 2.3 (1) Figure 1a is oversimplified and I think the authors should try to combine it with Figure S5 which is more precise as it describes collapse contigs.

Response 2.3: We agree and have updated Figure 1 with the suggested information.

Comment 2.4 (2) L106-107: "Reads aligned to diplotigs, triplotigs and tetraplotigs were randomly assigned to one of the respective haplotypes." In this case, even if the assembler had collapsed these regions, they may contain small variations between the haplotypes, and this random assignment can lead to mixing the haplotypes.

Response 2.4: This is correct. Therefore, we have searched all IBD blocks for "pseudo-heterozygous" differences using short read alignments (line 174-179; Supplementary Information: section "Identification of potentially collapsed variants within IBD blocks"; Data S7). In this analysis (which was originally described in the supplement of the first version of the manuscript), we found that the amount of collapsed variation was 1 SNP per 72-81 kb in diplotig, triplotig and tetraplotig regions. While we agree that the collapse of variation could be a potential weakness, the actual impact of it is extremely low. After all, if the degree of differences would be higher, the haplotypes could be eventually be resolved anyways.

Comment 2.5 (3) L115-116: the authors should explain in more detail why the LG10 is shorter in their assembly compared to existing assemblies.

Response 2.5: Five large-scale rearrangements distinguish the Chr.10 haplotypes of 'Otava' from the currently known haplotypes of Chr. 10 (Supplementary Information Fig.S10d; Fig. S20). Despite these differences, all four independently assembled haplotypes revealed the same five large-scale rearrangements which supports the correctness of these assemblies. Recent work by Luca Comai's lab reported large-scale rearrangements of similar size (<https://www.biorxiv.org/content/10.1101/2021.06.18.449059v2>) evidencing that such large differences do segregate in cultivated potato genomes (line 136-139). However, it will require the assembly of more haplotypes to describe the entire extent of such drastic haplotype differences and more analysis to understand how they interact if combined in one individual genome.

Comment 2.6 (4) The assembly has been polished with Pilon which performs an error correction by examining the pileup of bases from the reads. Unfortunately this strategy can lead to haplotype mixing, did the authors check the local accuracy?

Response 2.6: We agree that polishing relies on the correctness of the read alignments and that wrongly aligned reads can lead to errors. We addressed this with two rounds of polishing, i.e. after polishing with short reads, we polished the assembly again using the HiFi reads (Supplementary Information: section "Haplotype-specific PacBio HiFi read separation and haplotype assembly"). This second round of polishing was performed with reads from the individual haplotypes as we used them for the haplotype-specific assemblies which drastically reduced the possibility of wrong alignments.

To evaluate the sequence quality of the polished assembly, we have now added a k-mer based analysis using Merqury, which revealed an extremely high base accuracy with a QV of 51.7 (or around 6.7 errors per million bases) across the polished assembly (line 131-133; Supplementary Information: section "Evaluation of assembly quality").

To additionally evaluate the effects of the polishing on the haplotyping, we repeated the parental k-mer analysis with the polished contigs (Supplementary Information: section "Evaluation of haplotyping accuracy"). After polishing we observed the same haplotyping accuracy of 99.6% as compared to the accuracy before polishing, supporting that polishing had not affected haplotyping.

Comment 2.7 (5) On Figure2, the light and dark blue boxes are not easily identifiable.

Response 2.7: We have updated the coloring scheme of Fig 2.

Comment 2.8 (6) The authors report that a large fraction of the genes were not present in the four haplotypes. Could this result be an artefact of the annotation process? I read that the authors use the "--percent 60" option of exonerate but how Hisat and StringTie handled the presence of four haplotypes?

Response 2.8: *The gene annotation was performed for each haplotype sequence independently. This implies that Hisat and StringTie were used for each haplotype separately and did not have to handle the presence of four haplotypes. The results were then combined with alignments of homologous protein sequences (UnitProtKB) and ab initio gene predictions to determine final gene models (Supplementary Information Section "Genome annotation and assessment").*

Comment 2.9 More generally, I was a little surprised by this result, because the BUSCO scores are high in each of the four haplotypes. Does this mean that BUSCO gene sets are biased?

Response 2.9: *In fact, the BUSCO scores of the annotations of the individual haplotypes were actually much lower than the BUSCO score of the entire genome supporting the fact that a large fraction of genes is missing in individual haplotypes. The individual haplotypes featured BUSCO scores for single-copy genes of 88.6% to 90.4%, while the overall score for all four haplotypes was 97.3% (Supplementary Information Data S3). We have clarified this in the main text (line 241-249).*

Reviewer #3:

Remarks to the Author:

Review of the paper entitled "Chromosome-scale and haplotype-resolved genome assembly of a tetraploid potato cultivar" by Sun, H. et al. submitted to Nature Genetics.

This paper reports de novo assembly of four haplotypes of a tetraploid potato (*Solanum tuberosum*) cultivar. Potato is the most important root crop of the world. However, its autotetraploid genome and high levels of inbreeding depression hampered application of modern breeding techniques to the crop. Potato genome sequences have been reported previously using a di-haploid individual derived from a homozygous diploid potato (The Potato Genome Sequencing Consortium, 2011) and a heterozygous diploid potato (Zhou et al. 2020). In the latter study, haplotype-resolved genome assembly was reported using linkage information obtained by WGS of selfed F2 progeny.

The submitted paper is the first report of haplotype-resolved genome assembly of a cultivar Otava with a tetraploid genome, which comprises the majority of the cultivated potato. First, the authors obtained PacBio long reads and contigs. To obtain the four haplotypes, the authors employed Illumina WGS of gametic genomes of 717 pollens (following the approach of Shi et al. 2019) from the individual Otava. Gametic reads were mapped to the contigs, and presence/absence of reads in the contigs allowed grouping of the contigs to the four haplotypes. De novo assembly was conducted again for the reads belonging to each of the four groups, and the generated haplotypes were validated by Hi-C. Also, genome sequences of Stieglitz and Hera, the two parents of Otava, was sequenced to confirm the haplotypes.

Comparison of the four haplotypes revealed that 50% of the genomic regions were identical (IBD) in at least two haplotypes. On the other hand, higher divergence between the haplotypes were marked by large-scale structural changes involving inversion, duplication, translocation, suggesting introgression from wild species. Gene comparison among the four haplotypes showed that only 53.6% of the genes were present in all four haplotypes, and the rest were shared by a subset of haplotypes. Remarkably, the genes with four identical alleles show enrichment in GO of photosynthesis, chlorophyll binding and translation, suggesting selective optimization of the tetraploid potato.

By employing an innovative and solid genomics approach, this paper revealed genome structure of autopolyploid crop species for the first time. This study will be a benchmark study of genomics of polyploid species which are common in crops.

Thank you very much for pointing out the high relevance of our work.

Specific points:

Comment 3.1 1/ Figure S5, S6, S7, Figure 3b: Allocation of pollen reads to contigs to group them into four haplotypes for 12 chromosomes is overall clear. However, it is not clear how the authors solved the junction between the haplotype-unique regions and the shared regions. As an example, in Fig. S15, Hera_Chr_4_2 (Hera_2), Stie_Chr_4_2 (Stie_2), Stie_Chr_4_1 (Stie_1) share a ~30Mb trip-H1S1S2 IBD region (IBD) in the middle between the unique regions on the left (left) and right (right) sides of the chromosome. With the method described,

it seems possible that “Hera_2 left - IBD - Stie_1 right” or “Stie_1 left – IBD – Stie_2 right” configurations are also obtained. An explanation will benefit the understanding of readers.

Response 3.1: *Thank you for this comment. We agree that a detailed description on this is required as this is one of the key points that distinguishes our assembly method from “sequencing data only” assemblies. Such assemblies would fail in connecting regions next to long IBD blocks. In contrast, our method connects genomic regions using genetic linkage information inferred from single-cell genome analyses of a few hundred of pollen genomes. Even distantly located genomic regions of the same haplotype feature highly correlated (i.e. linked) genotypes while regions of the other haplotypes will not be linked at all. Therefore, it is possible to group all linked genomic regions (contigs) from the same haplotype.*

We have added additional explanations and an example on linkage grouping to the main text to make these points clear (Fig. 1c; line 115-119; Supplementary Information: section “Linkage-based grouping of contigs”).

Comment 3.2 2/ The paper does not mention recombination of haplotypes in the gametes. A description how it affects and not affect the method is needed.

Response 3.2: *We agree that explaining the impact of meiotic recombination on the reconstruction of the haplotypes helps in understanding the method. We have now added details on this to the main text stating that “recombination breakpoints, which are integrated into the pollen genomes during meiosis, change the haplotype within a pollen genome, and thereby slightly change the PAP [markers] along the chromosome. However, as recombination is generally rare, closely linked coverage markers still feature highly correlated PAPs” (line 115-119; Supplementary Information: section “Linkage-based grouping of contigs”).*

The grouping of the PAPs is therefore possible even in the presence of recombination breakpoints. In fact, this grouping is similar to the first steps during the calculation of a genetic map, where genomic regions, which are separated by some recombination events, can be connected through linkage.

Email (of 30th of September 2021) sent by the NCBI SRA Curator confirming that our data will be released soon:

From: "Seto, Charlie (NIH/NLM/NCBI) [C]" <charlie.seto@nih.gov>
Subject: RE: Data releases for PRJNA726019
Date: 30. September 2021 at 16:48:42 CEST
To: Hequan Sun <sun@mpipz.mpg.de>

Thanks for confirming!

Will do.

From: Hequan Sun <sun@mpipz.mpg.de>
Sent: Thursday, September 30, 2021 10:47 AM
To: Seto, Charlie (NIH/NLM/NCBI) [C] <charlie.seto@nih.gov>
Subject: Re: Data releases for PRJNA726019

Hello Seto,

we (=Saurabh: spophaly@mpipz.mpg.de and I) are working on the same data. Please kindly help release all SRA accessions under this BioProject: PRJNA726019.

Thank you very much!

Bes regards,
Hequan Sun

MPIPZ

On 30. Sep 2021, at 16:38, Seto, Charlie (NIH/NLM/NCBI) [C] <charlie.seto@nih.gov> wrote:

Hello,

I fielded a request for remaining releases from this submission from someone at your institution. However, they are not listed in your lab page; and I wanted to clarify **your** intentions, since you are data submitter.

We have a bioproject with accession PRJNA726019 which was made public on **2021-08-14** (available here <https://dataview.ncbi.nlm.nih.gov/object/PRJNA726019>). However not all submissions inside this bioproject have been made public and are marked with status "To be released". There is no option to change the release date as well (e.g <https://dataview.ncbi.nlm.nih.gov/object/SRR15198317>) . A few examples include [SRR15248702](https://dataview.ncbi.nlm.nih.gov/object/SRR15248702) , [SRR15248701](https://dataview.ncbi.nlm.nih.gov/object/SRR15248701) .

We would be grateful if all the associated data in this bioproject is released as soon as possible due to an urgent publication deadline.

I would be glad to provide more information if required.

Charlie Seto, NCBI, SRA Curator

Decision Letter, first revision:

Our ref: NG-LE57735R

16th Nov 2021

Dear Dr. Schneeberger,

Thank you for submitting your revised manuscript "Chromosome-scale and haplotype-resolved genome assembly of a tetraploid potato cultivar" (NG-LE57735R). It has now been seen by the original referees and their comments are below. The reviewers find that the paper has improved in revision, and therefore we'll be happy in principle to publish it in Nature Genetics, pending minor revisions to satisfy the referees' final requests and to comply with our editorial and formatting guidelines.

Sincerely,
Wei

Wei Li, PhD
Senior Editor
Nature Genetics
New York, NY 10004, USA
www.nature.com/ng

Reviewer #1 (Remarks to the Author):

The authors have amply addressed all of my previous comments. My only minor comment on the new revision is that the new sentences at the beginning of the Discussion (L291) are unclear. "The sequence differences between the haplotypes were much higher as compared to the differences commonly found within species and were rather reminiscent of the differences found between species." This seems vague and unscientific, as some species (eg maize) contain very high diversity and other genera contain different species by nomenclature that are very genomically similar. Do they mean related potato species? If so it might be better to specify exactly which species they are comparing the intra- and inter-species diversity to.

Reviewer #2 (Remarks to the Author):

I have read the revised manuscript "Chromosome-scale and haplotype-resolved genome assembly of a tetraploid potato cultivar" which presented in its previous version, the first haplotype-resolved genome assembly of a tetraploid species. The methods and results were considered to be of high-quality, but the lack of scientific findings was pointed out.

In the revised version the authors addressed my comments with in particular the provision of data. I liked the additional discussion regarding the low number of distinct alleles per gene.

First, the authors added an analysis of non-coding RNAs that was proposed by a referee, and in my opinion the results presented here are descriptive and therefore of little interest.

More interestingly, the authors generated new data and compared the gene expression and methylation in the four haplotypes. The highlighting of the 1,219 genes with allele-specific expression is interesting but I was a little frustrated that the authors did not inspect the function of these ASE genes in more detail. In Figure 4g (and Data S12), the correlation between ASE and methylation is strong on all 304 genes but I would expect to have correlation values for all 1,219 genes. Again, an examination of gene functions is relevant.

Reviewer #3 (Remarks to the Author):

Review of the revised paper entitled "Chromosome-scale and haplotype-resolved genome assembly of a tetraploid potato cultivar" by Sun, H. et al. submitted to Nature Genetics.

The authors appropriately addressed all the comments of this reviewer, which made the innovative method clearly understood by the readers. With the addition of RNA-seq data and methylome data, biological aspect of the paper was also substantially improved. I congratulate the authors for this benchmark study of haplotype-resolved sequencing of a polyploid crop genome. I believe the method developed here will be applied to many other polyploid species to understand their genome structures, genome evolution, gene dosage and gene expression.

Minor comment

L54: additional efforts such genetic maps -> additional efforts such as genetic maps?

Author Rebuttal, first revision:

Reviewers' Comments:

Reviewer #1:

Remarks to the Author:

Comment The authors have amply addressed all of my previous comments. My only minor comment on the new revision is that the new sentences at the beginning of the Discussion (L291) are unclear. "The sequence differences between the haplotypes were much higher as compared to the differences commonly found within species and were rather reminiscent of the differences found between species." This seems vague and unscientific, as some species (eg maize) contain very high diversity and other genera contain different species by nomenclature that are very genomically similar. Do they mean related potato species? If so it might be better to specify exactly which species they are comparing the intra- and inter-species diversity to.

Response: *Thank you for this comment, we have now changed this paragraph pointing out that the genomes of many potato cultivars include introgressions from wild species as it also can be observed in the pedigree of 'Otava'. In its pedigree, for example, we can find 'Edinense fraglich' (or EF), which was probably a variety of *S. demissum* or a hybrid of *S. demissum* and *S. edinense* or *S. tuberosum*, and was used to introduce resistance against *Phytophthora infestans*.*

Reviewer #2:

Remarks to the Author:

I have read the revised manuscript "Chromosome-scale and haplotype-resolved genome assembly of a tetraploid potato cultivar" which presented in its previous version, the first haplotype-resolved genome assembly of a tetraploid species. The methods and results were considered to be of high-quality, but the lack of scientific findings was pointed out. In the revised version the authors addressed my comments with in particular the provision of data. I liked the additional discussion regarding the low number of distinct alleles per gene. First, the authors added an analysis of non-coding RNAs that was proposed by a referee, and in my opinion the results presented here are descriptive and therefore of little interest. More interestingly, the authors generated new data and compared the gene expression and methylation in the four haplotypes.

Comment 2.1 The highlighting of the 1,219 genes with allele-specific expression is interesting but I was a little frustrated that the authors did not inspect the function of these ASE genes in more detail.

Response 2.1: *We have performed a functional analysis of the 1,219 genes, and the results showed that the genes were enriched in hydrolase activity, photosynthesis, light harvesting and RNA methylation, which is consistent with the "GO enrichment analysis of genes with four identical alleles" which include many photosynthesis-related genes (Fig. 4c; Supplementary Table 12).*

Comment 2.2 In Figure 4g (and Data S12), the correlation between ASE and methylation is strong on all 304 genes but I would expect to have correlation values for all 1,219 genes. Again, an examination of gene functions is relevant.

Response 2.2: *We have repeated the analysis by calculating the correlations based on absolute methylation values and expression values, and now found 327 genes with a significant correlation between expression and methylation. We now present this (updated) correlation analysis for all 1,219 genes (Fig. 4g; Supplementary Table 13 and 14, Supplementary Figure 11). In addition, we performed GO analyses on each of the (correlated and non-correlated) gene sets. However, no significantly enriched gene function was found among the genes in each of these sets (in addition to the enrichment found among the 1,219 genes).*

Reviewer #3:

Remarks to the Author:

Review of the revised paper entitled "Chromosome-scale and haplotype-resolved genome assembly of a tetraploid potato cultivar" by Sun, H. et al. submitted to Nature Genetics.

The authors appropriately addressed all the comments of this reviewer, which made the innovative method clearly understood by the readers. With the addition of RNA-seq data and methylome data, biological aspect of the paper was also substantially improved. I congratulate the authors for this benchmark study of haplotype-resolved sequencing of a polyploid crop genome. I believe the method developed here will be applied to many other polyploid species to understand their genome structures, genome evolution, gene dosage and gene expression.

Specific points:

Comment Minor comment L54: additional efforts such genetic maps -> additional efforts such as genetic maps?

Response: *We have corrected the text accordingly.*

Final Decision Letter:

In reply please quote: NG-A57735R1 Schneeberger

10th Jan 2022

Dear Dr. Schneeberger,

I am delighted to say that your manuscript "Chromosome-scale and haplotype-resolved genome assembly of a tetraploid potato cultivar" has been accepted for publication in an upcoming issue of Nature Genetics.

Your paper will be published online after we receive your corrections and will appear in print in the next available issue. You can find out your date of online publication by contacting the Nature Press Office (press@nature.com) after sending your e-proof corrections. Now is the time to inform your Public Relations or Press Office about your paper, as they might be interested in promoting its publication. This will allow them time to prepare an accurate and satisfactory press release. Include your manuscript tracking number (NG-A57735R1) and the name of the journal, which they will need

when they contact our Press Office.

Please note that *Nature Genetics* is a Transformative Journal (TJ). Authors may publish their research with us through the traditional subscription access route or make their paper immediately open access through payment of an article-processing charge (APC). Authors will not be required to make a final decision about access to their article until it has been accepted. [Find out more about Transformative Journals](https://www.springernature.com/gp/open-research/transformative-journals)

Authors may need to take specific actions to achieve compliance with funder and institutional open access mandates. For submissions from January 2021, if your research is supported by a funder that requires immediate open access (e.g. according to [Plan S principles](https://www.springernature.com/gp/open-research/plan-s-compliance)) then you should select the gold OA route, and we will direct you to the compliant route where possible. For authors selecting the subscription publication route our standard licensing terms will need to be accepted, including our [self-archiving policies](https://www.springernature.com/gp/open-research/policies/journal-policies). Those standard licensing terms will supersede any other terms that the author or any third party may assert apply to any version of the manuscript.

Please note that Nature Research offers an immediate open access option only for papers that were first submitted after 1 January, 2021.

If you have not already done so, we invite you to upload the step-by-step protocols used in this manuscript to the Protocols Exchange, part of our on-line web resource, natureprotocols.com. If you complete the upload by the time you receive your manuscript proofs, we can insert links in your article that lead directly to the protocol details. Your protocol will be made freely available upon publication of your paper. By participating in natureprotocols.com, you are enabling researchers to more readily reproduce or adapt the methodology you use. [Natureprotocols.com](http://natureprotocols.com) is fully searchable, providing your protocols and paper with increased utility and visibility. Please submit your protocol to <https://protocolexchange.researchsquare.com/>. After entering your nature.com username and password you will need to enter your manuscript number (NG-A57735R1). Further information can be found at <https://www.nature.com/nprot/>.

Sincerely,

Wei Li, PhD
Senior Editor
Nature Genetics
New York, NY 10004, USA
www.nature.com/ng